# Microstructural and functional plasticity following repeated brain stimulation during cognitive training in older adults

Daria Antonenko ®[1] ✉, Anna Elisabeth Fromm[1], Friederike Thams ®[1], Ulrike Grittner ®[2,3], Marcus Meinzer[1] & Agnes Flöel[1,4]

The combination of repeated behavioral training with transcranial direct current stimulation (tDCS) holds promise to exert beneficial effects on brain function beyond the trained task. However, little is known about the underlying mechanisms. We performed a monocenter, single-blind randomized, placebo-controlled trial comparing cognitive training to concurrent anodal tDCS (target intervention) with cognitive training to concurrent sham tDCS (control intervention), registered at ClinicalTrial.gov (Identifier NCT03838211). The primary outcome (performance in trained task) and secondary behavioral outcomes (performance on transfer tasks) were reported elsewhere. Here, underlying mechanisms were addressed by pre-specified analyses of multimodal magnetic resonance imaging before and after a three-week executive function training with prefrontal anodal tDCS in 48 older adults. Results demonstrate that training combined with active tDCS modulated prefrontal white matter microstructure which predicted individual transfer task performance gain. Training-plus-tDCS also resulted in microstructural grey matter alterations at the stimulation site, and increased prefrontal functional connectivity. We provide insight into the mechanisms underlying neuromodulatory interventions, suggesting tDCS-induced changes in fiber organization and myelin formation, glia-related and synaptic processes in the target region, and synchronization within targeted functional networks. These findings advance the mechanistic understanding of neural tDCS effects, thereby contributing to more targeted neural network modulation in future experimental and translation tDCS applications.

Developing effective cognitive interventions to reduce or even prevent age-associated brain impairment has received substantial scientific attention in aging societies worldwide. Preliminary evidence suggests that the combination of behavioral training and concurrent transcranial electrical stimulation (tES), one of the most widely used noninvasive brain stimulation (NIBS) techniques, may induce cross-task cognitive benefits, in young adults and advanced age[1–6]. In particular, repeated sessions of one variant of tES, anodal transcranial direct current stimulation (tDCS), with cognitive training can boost training gains, with the potential to induce cognitive enhancement lasting up to one month[5,6]. For instance, anodal tDCS over dorsolateral prefrontal cortex during executive training resulted in enhanced working

[1]Department of Neurology, Universitätsmedizin Greifswald, Greifswald, Germany. [2]Berlin Institute of Health (BIH), Berlin, Germany. [3]Charité – Universitätsmedizin Berlin, Humboldt-Universität zu Berlin, Berlin Institute of Health, Institute of Biometry and Clinical Epidemiology, Berlin, Germany. [4]German Centre for Neurodegenerative Diseases (DZNE) Standort Greifswald, Greifswald, Germany. ✉e-mail: daria.antonenko@med.uni-greifswald.de

memory performance in anodal compared to sham groups in trained or near-transfer tasks[3,7–9]. However, evidence on beneficial effects is still not unequivocal[10–12], and add-on effects are often small and variable between individuals depending on internal or external factors[2]. Therefore, a better understanding of the underlying mechanisms by which tDCS exerts its beneficial effects in aging brains is of utmost importance to advance the potential of this technique.

As for learning-related brain plasticity, previous work has shown that the brain's microstructure can be modified by learning. Seminal work in post-mortem monkey brains showed that learning of a new skill indeed induces generation of denser and more extensive white matter projections[13,14]. In vivo visualization of learning-induced structural plasticity in both animals and humans is possible with diffusion tensor imaging (DTI)[15,16]. Main parameters from DTI sensitive to microstructural changes are fractional anisotropy (FA) and mean diffusivity (MD) with FA in white matter pathways reflecting directional coherence of fibers and MD in gray matter reflecting magnitude of water molecule diffusion[17]. Complementary histological analyses showed that, at the cellular level, changes in neural and non-neural dependent activity (e.g., synaptogenesis and changes in dendritic spine morphology) and changes in white matter (e.g., variation of axon diameter, myelin, packing density, fiber geometry) contribute to the observed alterations in neuroimaging data[15,16,18,19]. For instance, using DTI, Scholz et al. showed that skill training over several weeks induced changes in white matter properties in humans, potentially reflecting changes of myelin, or altered packing density[20]. Similar microstructural remodeling processes following learning, documented by changes in DTI parameters, in both white and gray matter structures have been demonstrated in rodent and human brains[15,19,21,22]. In sum, while microstructural changes, assessed by DTI, have been demonstrated in several studies to result from training, their exact timescale, the contributing cellular processes, and their relationship to individual learning magnitudes are yet not completely understood[15,20].

TES non-invasively modulates excitability and synaptic plasticity in neurons[2]. Repeated tDCS sessions can induce long-lasting changes in excitability and synaptic efficiency, inducing long-term potentiation (LTP)-like effects[23–25]. A safe and commonly used range of tDCS dose is 1–2 mA for up to 30 min[26]. Previous evidence from proof-of-concept studies suggests general efficacy of applying anodal tDCS with 1 mA for 20 min in single and repeated stimulation sessions, in young and in older adults[5,6,27]. Importantly, titration studies systematically comparing different stimulation parameters have shown non-linearity of intensity-dependent neuroplastic effects[28,29]. In older compared to young adults, higher intensities may be necessary to induce behavioral changes[30,31], given that less current may reach the brain due to age-related atrophy which reduces electric fields[32].

Simultaneous tDCS-fMRI application in proof-of-principle studies revealed changes in local activity and functional connectivity (temporally coherent activity between brain regions) that predicted behavioral performance gains[33–35]. Functional connectivity modulations in response to repeated training sessions may even reflect network level reorganizations, promoting longer-lasting neural plasticity[4,36–38].

Establishing the underlying cellular (and molecular) mechanisms in the human brain can advance understanding of neuromodulatory plasticity. Animal models suggest modified tissue density due to altered neuronal morphology (e.g., size/shape of axons, dendritic spines and cell bodies), altered glial cells activity or reorganization/reshaping of synaptic connections as neuroplastic phenomena induced by tDCS[24,39]. Combined repeated tDCS-plus-training interventions in human participants which promote plasticity have been suggested to induce microstructural changes in brain white and gray matter similar to those induced by learning, but direct evidence is limited[25]. Importantly, multimodal imaging is necessary to establish a comprehensive understanding of the underlying neurophysiological mechanisms[40].

In this work, we tested the hypotheses that concurrent anodal prefrontal tDCS administered across repeated cognitive training sessions would modulate white matter microstructure in cortical target areas and associated neural networks compared to training with placebo (sham) stimulation. tDCS (1 mA) was administered for 20 min concurrently with two executive function training tasks (letter updating training, decision-making). While there were no between-group differences in the primary outcome (performance on letter-updating), we observed superior near-transfer effects (performance on N-back) in the tDCS group at post-intervention and follow-up, but in no other transfer tasks (please see ref. 41 for the behavioral results of the study). In the current paper, we used DTI acquired before and after the intervention for individual fiber tractography and quantification of white matter microstructure. Further, DTI allowed us to examine whether microstructural properties in the stimulation target would change due to the intervention as suggested previously[38]. The investigation of microstructural plasticity markers was complemented by resting-state functional magnetic resonance imaging (rs-fMRI) to analyze functional synchrony modifications within the targeted (frontoparietal) network. In order to explore the behavioral relevance of neural alterations, we further performed correlational analyses with LU (training, primary behavioral outcome) and N-back (corresponding near-transfer task with enhanced performance in the target compared to the control intervention)[41].

## Results

We performed a monocenter, single-blind randomized, placebo-controlled trial comparing cognitive training to concurrent anodal tDCS (target intervention) with cognitive training to concurrent sham tDCS (control intervention), registered at ClinicalTrial.gov (Identifier NCT03838211). The study was conducted from February 15, 2018 (first participant enrolment) to March 25, 2020 (last participant enrolment). The primary outcome (performance in trained task) and secondary behavioral outcomes (performance on transfer tasks) were reported elsewhere. Here, underlying mechanisms were addressed by pre-specified analyses of multimodal magnetic resonance imaging before and after an executive function training with prefrontal anodal tDCS in 48 older adults[41,42]. Participants were randomly assigned to two groups (anodal and sham tDCS) using stratified blockwise randomization (based on age and baseline performance). All participated in three weekly training sessions provided over three weeks (nine sessions total). The training comprised letter updating and decision-making tasks, lasting ~40 min. TDCS was applied daily with an intensity of 1 mA for 20 min (30 s for sham group), starting briefly prior to the first training task. A conventional tDCS montage was used that targeted prefrontal functions[43]. The anode was centered over the left dorsolateral prefrontal cortex (F3 of the 10–20 EEG system; size: 5 cm diameter); the cathode over the contralateral supraorbital area (Fp2; size: 5 cm diameter). Incidence of adverse events did not differ between groups (incidence rate ratio [95% CI]: 0.8 [0.4, 1.9], Supplementary Table 1) and the James Blinding index (mean [95% CI]: 0.67 [0.55, 0.80], Supplementary Table 2) indicated blinding success (see Supplementary Methods and Results for more information). MRI was performed two days prior to and two days after the intervention (Table 1, Fig. 1). MRI comprised different imaging modalities to investigate effects on neural networks (Methods), such as DTI to quantify structural plasticity due to the intervention in white matter tracts using individual probabilistic tractography (defined as fractional anisotropy, FA, which reflects directional preference of diffusion) as well as in gray matter microstructure of the stimulation target (defined as mean diffusivity, MD, which reflects molecular diffusion rate) and resting-state functional resonance imaging (rs-fMRI) to examine alterations in functional network connectivity (FC, defined as temporal correlation of blood-oxygenation level-dependent, BOLD, signals in areas of the network). Common software pipelines were used for MR data analyses. We

## Table 1 | Demographic characteristics

|  | Total | Anodal | Sham |
|---|---|---|---|
| N (n females) | 48 (31) | 22 (14) | 26 (17) |
| Age [years] | 69.8 (3.9) | 69.3 (4.3) | 70.2 (3.7) |
| Education [years] | 15.5 (2.2) | 16.0 (2.0) | 15.2 (2.3) |
| APOE e4 [N] | 12 | 6 | 6 |
| Depression [GDS score] | 1.1 (1.2) | 1.1 (1.1) | 1.1 (1.3) |
| CERAD [Total score] | 88.4 (4.8) | 89.5 (4.2) | 87.4 (5.1) |

Mean and SD values (except for "N") are provided.
Total score (max. 100) based on[114] with components from verbal fluency, Boston Naming Test, constructional praxis, word list learning, word list recall, word list recognition. Source data are provided as a Source Data file.
*APOE* Apolipoprotein E, *GDS* Geriatric depression scale (max. score: 15 with a cut of off of 6)[113].
*CERAD* Consortium to Establish a Registry for Alzheimer's Disease.

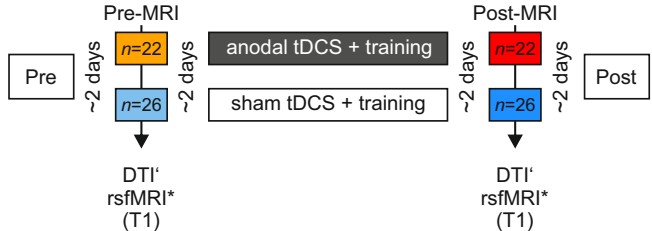

**Fig. 1 | Study flow chart.** Following a pre-assessment of performance on the cognitive tasks, a pre-intervention MRI was conducted; the intervention commenced two days later and lasted for three weeks (with active (anodal) or sham tDCS + training administered three times per week). A post-intervention MRI session was conducted two days after the end of the intervention period. MRI magnetic resonance imaging, tDCS transcranial direct current stimulation. 'Exclusion of *n* = 2 (one from anodal in post, one from sham group in pre) in tractography analysis due to missing DTI data. *Exclusion of *n* = 1 from sham group resting-state fMRI analysis due to exessive motion during the functional scan.

investigated the differences of FA, MD, and FC between anodal and sham groups after the intervention to scrutinize potential add-on effects of anodal tDCS during cognitive training. We also explored linear relationships between the effects on different MRI markers, and between MRI markers and performance gain in working memory (i.e., LU and N-back task).

### White matter microstructure is modulated after brain stimulation

We performed individual probabilistic tractography seeding from the stimulation target (left middle frontal gyrus, defined to represent the gyrus below the anodal electrode, picked from the Harvard-Oxford atlas[44]) using pipelines from FSL[45] to delineate prefrontal white matter pathways. This method repeatedly samples the distribution at each voxel to produce 'streamlines' that connect voxels from the selected seed region. To quantify microstructural integrity in these pathways before and after the intervention, FA values were extracted (averaging individual voxel values along the tract) for both time points and groups and entered into linear model analyses with values post intervention as dependent variables and group as between-subjects factor (including pre intervention FA values, age, and sex as covariates). We observed a group difference with higher FA values along the tract in the anodal compared to the sham group post-intervention ($t_{41} = -2.607$, $p = 0.013$, partial $\eta^2 = 0.14$; model-derived adjusted estimated means [CI]: 0.348 [0.343, 0.354] for anodal and 0.339 [0.334, 0.344] for sham group, Fig. 2). Pre FA values were positively associated with higher post FA values ($t_{41} = 10.343$, $p < 0.001$, partial $\eta^2 = 0.72$). No interaction of pre-training FA values with stimulation group was observed and therefore no interaction term was included in the final model. No substantial

associations of age and sex to post FA were observed ($t$'s < 1.22, $p$'s > 0.23, partial $\eta^2$'s < 0.05). Tract volumes did not change through the intervention ($t_{41} = 0.547$, $p = 0.587$, partial $\eta^2 = 0.01$; model-derived adjusted estimated means [CI]: 3905 [3669, 4141] for anodal and 3814 [3563, 4065] for sham group). In sum, FA within the structural target network increased more in the training group that had received anodal tDCS compared to sham tDCS, suggesting that active tDCS-plus-training modulated white matter tract microstructure.

To evaluate the robustness of FA results, we conducted complementary tract-based spatial statistics (TBSS)[46] and automated global tractography analyses with anatomical priors (using tracts constrained by underlying anatomy, TRACULA)[47]. TBSS creates a mean skeleton, representing the centres of all tracts common to the group. Each participants' FA data is then projected onto this skeleton. Whole-brain voxel-wise statistical comparisons showed significant relative FA increases in anodal compared to sham group in left and right lateral prefrontal, medial prefrontal and parietal regions (permutation test, $p < 0.05$, TFCE-corrected, see Supplementary Table 3 and Supplementary Fig. 1). To delineate specific fiber systems, we overlaid the canonical pathway from the individual probabilistic tractography with atlas labels of the John's Hopkins University (JHU) white matter atlas (Supplementary Fig. 2). The resultant two tracts of interest (i.e., prefrontal section of the body of the corpus callosum, CC; part of the left superior longitudinal fasciculus, SLF) were then reconstructed using TRACULA. FA values in the CC were higher in the anodal compared to the sham group (main effects $t_{40} = -1.96$, $p = 0.058$, partial $\eta^2 = 0.09$) and an interaction of initial FA values by group was found ($t_{40} = 2.01$, $p = 0.051$, partial $\eta^2 = 0.09$). Thus, beneficial stimulation effects were larger for individuals with higher FA at baseline (e.g., for low baseline values at 25th percentile (0.52), anodal: 0.52 [0.51, 0.53], sham: 0.52 [0.51, 0.53], $p = 0.530$; for high baseline values at 75th percentile (0.57), anodal: 0.57 [0.56, 0.58], sham: 0.56 [0.55, 0.57], $p = 0.089$). FA in the SLF did not change through the intervention ($t_{41} = 0.02$, $p = 0.984$, partial $\eta^2 = 9.9e{-}06$; model-derived estimated means [CI]: 0.42 [0.41, 0.42] for anodal and 0.42 [0.41, 0.42] for sham group). In sum, we found increased FA values in the CC (prefrontal section of the body) in anodal compared to sham for individuals with higher baseline FA while no difference was observed for the left SLF (see Supplementary Methods and Results for further details).

### Gray matter microstructure is altered after brain stimulation

Gray matter regions in the cortex underneath the anode (left middle frontal cortex) were segmented using Freesurfer[48], overlaid on the stimulation target (Fig. 3) and projected into DTI space to extract individual MD values before and after the intervention. MD values were entered into linear model analyses with values post intervention as dependent variables and group as between-subjects factor (including covariates pre intervention MD, age, and sex). MD values after the intervention were lower in the anodal compared to sham tDCS groups (main effects $t_{41} = -2.30$, $p = 0.027$, partial $\eta^2 = 0.11$) and an interaction of initial MD values by group was found ($t_{41} = 2.29$, $p = 0.027$, partial $\eta^2 = 0.11$). Thus beneficial stimulation effects were larger for individuals with low MD at baseline (e.g., for low baseline values at 25th percentile $(0.9 \times 10^{-3})$, anodal: $0.9 \times 10^{-3}$ [0.8, $1.03 \times 10^{-3}$], sham: $1.07 \times 10^{-3}$ [1.00, $1.16 \times 10^{-3}$]; in contrast to those with high baseline values at 75th percentile $(1.15 \times 10^{-3})$, anodal: $1.18 \times 10^{-3}$ [$1.13, 1.24 \times 10^{-3}$], sham: $1.13 \times 10^{-3}$ [1.09, $1.18 \times 10^{-3}$]. Control analyses examined whether macrostructural changes may potentially explain microstructural differences. Therefore, stimulation effects on gray matter volume of middle frontal gyri was evaluated and showed no substantial effect ($t_{42} = 0.110$, $p = 0.913$, partial $\eta^2 < 0.01$, model-derived adjusted estimated means [CI]: 13,924 [13,764, 14,084] for anodal and 13,916 [13,759, 14,073] for sham group). In sum, MD was decreased through training-plus-tDCS in the gray matter underlying the stimulation target, suggesting changes in microstructure following the intervention.

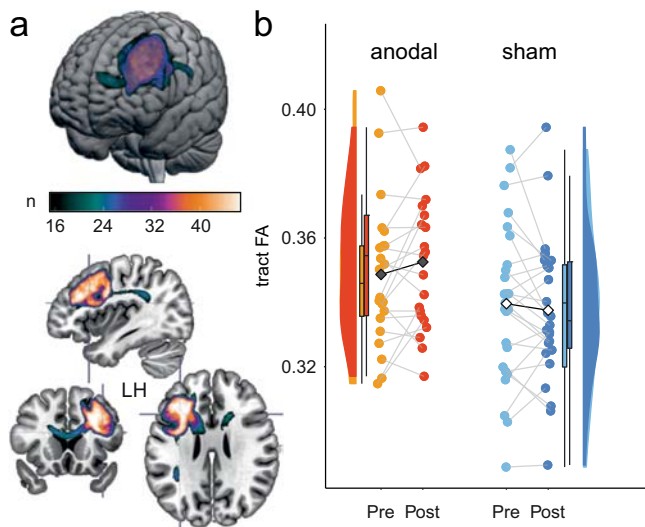

**Fig. 2 | White matter pathways' microstructure. a** A canonical image of the thresholded probabilistic tract, overlaid on the Montreal Neurological Institute (MNI) brain, created with MRIcroGL (https://www.nitrc.org/projects/mricrogl). To generate the canonical image, individual tracts from all participants were normalized, converted to binary images and then summed (color coding reflects the probability of voxels to be present in 33–100% of the participants). **b** Means (black diamonds for anodal and white diamonds for sham) and individual data points (single circles in orange/red for anodal and lightblue/darkblue for sham). Box plots indicate median (middle line), 25th, 75th (box), and 5th and 95th percentile (whiskers). *n* = 46 independent participants. FA along the tracts was increased after training in training group that had received anodal compared to sham tDCS. FA, fractional anisotropy. Source data are provided as a Source Data file.

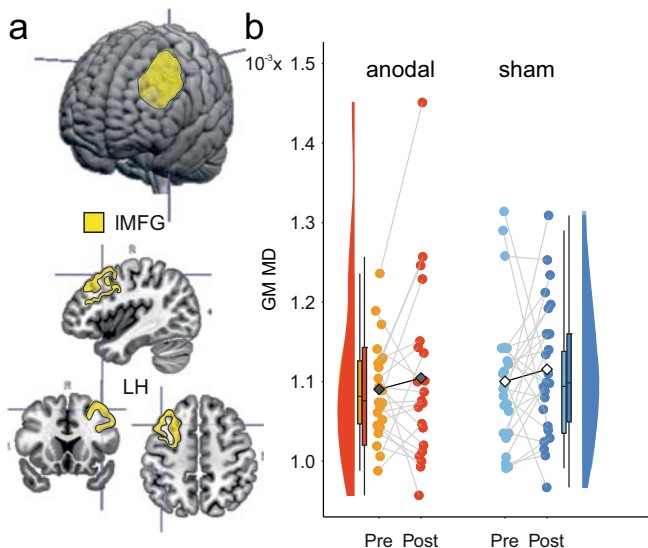

**Fig. 3 | Gray matter microstructure in the stimulation target. a** The left middle frontal gyrus (yellow), selected as the (gray matter) stimulation target, overlaid on the MNI brain, created with MRIcroGL (https://www.nitrc.org/projects/mricrogl), is provided on the left. **b** Means (black diamonds for anodal and white diamonds for sham) and individual data points (single circles in orange/red for anodal and lightblue/darkblue for sham). Box plots indicate median (middle line), 25th, 75th (box), and 5th and 95th percentile (whiskers). *n* = 46 independent participants. MD values were decreased after the intervention in anodal compared to sham group for those individuals with initially lower MD in the stimulation target. lMFG left middle frontal gyrus, LH left hemisphere, GM gray matter, MD mean diffusivity. Source data are provided as a Source Data file.

## Functional connectivity is increased after brain stimulation

To investigate whether functional connectivity was modulated by anodal tDCS, we performed seed-to-voxel correlational analyses on resting-state fMRI data using CONN[49]. The seed was selected to represent the area under the anode (left middle frontal gyrus from the Harvard-Oxford atlas, centered over F3[44], consistent with other tDCS studies using ROI approaches that demonstrated neural effects[35,38,50–55]) and Pearson's r correlation of the BOLD time course of this seed was computed across the entire brain. Subsequent second-level general linear model analysis for the group (anodal, sham) × time contrast (pre, post) revealed a significant cluster in the right prefrontal cortex (MNI coordinates: $x = 18$, $y = 18$, $z = 60$, $|T_{43}| > 3.53$, $k = 116$, $p < 0.05$ cluster-size FDR corrected p, voxel threshold: $p < 0.001$ p-uncorrected, adjusted for the covariates age and sex) in the right superior frontal gyrus (Fig. 4). A more liberal uncorrected p-threshold further supported that the cluster (covering in the right superior and middle frontal gyri) most likely reflects connectivity within the frontoparietal executive control network (MNI coordinates: $x = 18/32$, $y = 18/-4$, $z = 60/44$, $|T_{43}| > 2.96$, $k = 705/145$, $p < 0.05$ cluster-size p-FDR corrected, voxel threshold: $p < 0.005$ p-uncorrected, adjusted for the covariates age and sex). In sum, FC in the frontal-parietal network increased after training-plus-tDCS, suggesting enhanced network synchronization.

## Pathways' microstructure change is associated with performance gain

In order to explore linear relationships between the effects on different MR markers as well as with performance gain (LU and N-back change), correlation matrices were generated, illustrating scatterplots and Spearman correlation coefficients for all bivariate associations (Fig. 5). We observed a positive association between FA change and N-back change only, reflecting that individuals with higher increases in FA due to tDCS-plus-training also showed more pronounced

performance gains in the near transfer task ($r_S$ = 0.402, $p$ = 0.006, anodal: $r_S$ = 0.436, $p$ = 0.054, sham: $r_S$ = 0.251, $p$ = 0.23). Neither MD nor FC change showed an association with N-back change (|$r$|'s <0.299, $p$'s <0.15). A linear model, including all three levels of neural modulation with one model corroborated the results of the unadjusted correlational analyses (Supplementary Table 4). LU change which is more directly linked to the actual brain stimulation intervention (i.e., task networks directly targeted by tDCS), showed a positive association with FC change in the anodal tDCS group ($r_S$ = 0.420, $p$ = 0.046).

Bivariate scatterplots also revealed that microstructural plasticity in the stimulation target was associated with functional connectivity modulation: Higher decreases in gray matter MD were associated with increases in FC due to the intervention ($r_S$ = −0.336, $p$ = 0.022) with this relationship being more pronounced in the sham group ($r_S$ = −0.589, $p$ = 0.002) than in the anodal group ($r_S$ = −0.009, $p$ = 0.97), indicating that individuals with decreased MD showed increased prefrontal FC due to training.

As a control, we explored bivariate monotonic relationships between baseline integrity values (FA in white matter, MD in gray matter, and FC between the target and the resultant cluster) and behavioral performance gain. No substantial associations emerged (Supplementary Fig. 4). Thus, our results suggest a particular association of plasticity in white matter tracts with performance gain of the training-plus-tDCS intervention.

## Discussion

Three-week brain stimulation-assisted cognitive training in healthy older adults resulted in modifications of microstructure in white matter pathways and gray matter cortical target area as well as functional connectivity changes in a broader frontoparietal network. FA in prefrontal tracts originating from the stimulation target was increased in the group that had received anodal vs. sham tDCS, indicating higher integrity (i.e., directional preference of diffusion/directional

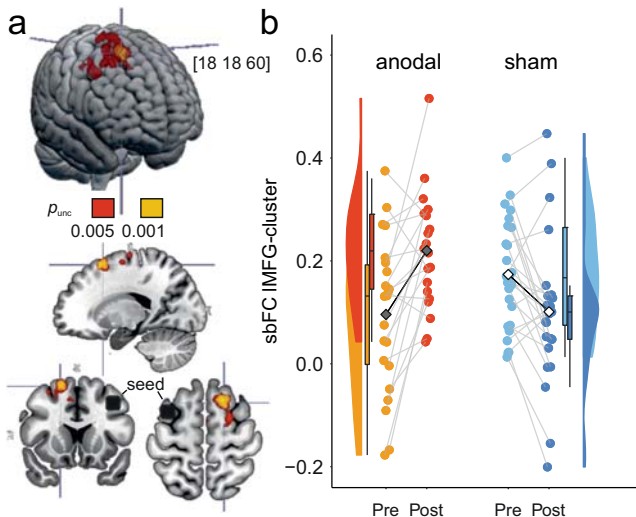

**Fig. 4 | Seed-based functional connectivity. a** Resultant cluster (red-yellow; $p_{FDR} < 0.05$, $p_{unc} < 0.001$) from seed-to-voxel resting-state functional connectivity analysis with seed in stimulation (black circle). Cluster location in the right superior and middle frontal gyri: increase of FC to the stimulation target after the intervention in anodal compared to sham group. Coordinates of the peak voxel are given [$x = 18, y = 18, z = 60$]. Brain images were created with MRIcroGL (https://www.nitrc.org/projects/mricrogl). **b** Means (black diamonds for anodal and white diamonds for sham) and individual data points (single circles in orange/red for anodal and lightblue/darkblue for sham). Box plots indicate median (middle line), 25th, 75th (box), and 5th and 95th percentile (whiskers). $n = 47$ independent participants. sbFC, seed-based functional connectivity. lMFG left middle frontal gyrus. Source data are provided as a Source Data file.

coherence) of frontoparietal white matter tracts which was associated with higher (transfer task) performance gains following the intervention. Further, gray matter microstructure changes differed between the stimulation groups, mainly in individuals with higher microstructural integrity (i.e., molecular diffusion rate/magnitude of water molecule diffusion) at baseline that showed decreased MD values after anodal vs. sham tDCS. Increased resting-state FC between prefrontal areas indicated additional synchronization within frontoparietal networks induced by tDCS. Overall, we provide evidence for microstructural and network modifications through brain stimulation in the human brain, which may characterize the underlying mechanisms of functional benefits due to the intervention.

We reconstructed individual tracts originating from the stimulation target in the left prefrontal cortex. Canonical images across our group of participants suggested that white matter fibers project from the stimulation target towards ipsilateral parietal and contralateral prefrontal areas[56], showing individual differences in their specific trajectories. FA along these tracts, reflecting microstructural integrity, was increased in the anodal compared to the sham group after the combined intervention. Complementary TBSS and TRACULA analyses confirmed the findings of individual probabilistic tractography seeding from the stimulation target (below the anodal electrode). Further, they suggested that the combined tDCS-plus-training effect may be rather promoted by transcallosal rather than ipsilateral frontoposterior pathways (i.e., interaction with pre FA values for the CC).

The DTI-derived index FA reflects the directional preference of diffusion and can be used to quantify the integrity of fiber organization in the human brain with higher values describing higher integrity[17]. Variability between individuals in white matter pathways mediating certain cognitive functions has been shown to predict the variability in behavioral performance[57]. Most intriguingly, these DTI metrics were used to delineate brain plasticity in vivo with their alterations being liked to long-term potentiation (LTP)[19,58]. Thus, cellular modifications through neuromodulatory interventions such as the density, myelin,

among others, which are indicative of LTP induction can be studied. Previous studies have observed changes in FA as a consequence of training, that were associated with behavioral performance gain[20,59], and even within short periods of time following learning[22]. For example, Hofstetter and colleagues found FA changes in the fornix, induced by a short-term (2 h) spatial training, providing evidence for rapid structural remodeling due to new learning experience. Our previous pilot study examining 3-day spatial training in older adults corroborated these initial results, suggesting that the behavioral relevance of dynamic remodeling in white matter tracts (rather than baseline microstructural integrity per se) is preserved in the aged brain[59].

In a seminal study investigating structural changes induced by a combined tDCS-and-physical therapy intervention in stroke patients, Zheng and Schlaug observed increased FA in descending motor tracts in the treatment but not in the control group[60]. However, as the control group did not receive any training, the results did not allow conclusions about whether effects were due to tDCS or training or both. Applying anodal tDCS over the left somatosensory cortex during repeated sensory learning, Hirtz and colleagues[61] found FA increases in anodal compared to sham group in the left frontal cortex, in the vicinity of the middle and superior frontal gyrus. The authors concluded that sensory learning involved prefrontal areas rather than stimulation target regions underneath the anodal electrode due to involvement of decision-making processes recruiting the frontoparietal network. In fact, these results together with our complementary findings of tDCS-induced microstructural plasticity in individual tracts may suggest a general (across domains) susceptibility of prefrontal white matter to tDCS-induced neuromodulation.

Candidate cellular mechanisms reflected in FA variations include alterations in cell membrane and fiber density, fiber coherence, axon diameter, myelination, collateral sprouting. While intracellular directional coherence contributes to the FA metric, extracellular properties have been shown to affect the diffusion of water molecules as well[16,17,62,63]. Given previous evidence, one possibility is that tDCS may affect fiber organization and myelin formation through rapid structural remodeling in white matter pathways originating from the stimulation target[60,64]. These myelination changes would then affect the speed of information processing between brain regions, underlying improvements of performance[20,65]. Other hypotheses have to be considered though, such as a potential effect of tDCS on tortuosity in the extracellular space, inducing differential changes in volume fractions in experimental groups (affecting water molecule motion and, as a consequence, the FA values)[17,66]. Future methodological research is needed to disentangle the contribution of these potential mechanisms to the observed tDCS-induced changes[17,24]. Importantly, the positive correlation of microstructural alterations with behavioral performance gain (as indicated by the transfer N-back task) may point towards a functional significance of preserved (brain stimulation-related and learning-related) neuromodulatory plasticity[19,22].

No correlation was observed for microstructural alterations with behavioral performance gain in the training (LU) task. Differences in the task related to the procedure of administration (e.g., repeated vs. single sessions), content (e.g. letters vs. numbers) and involved executive processes (continuous updating in conjunction with memorizing temporal order vs. active comparison operations) may not only affect brain activation patterns and, thus, magnitudes of behavioral modulation, but also the relationships to neural plasticity[67,68].

In order to examine microstructural changes within the gray matter of the stimulation target, MD values were extracted. A between-group comparison revealed an interaction between baseline MD values and the stimulation group effect, indicating a decrease after the intervention in the anodal compared to the sham group for individuals with initially lower values in the stimulation target.

The DTI-derived index MD reflects the molecular diffusion rate and is used to quantify tissue microstructure. Higher MD

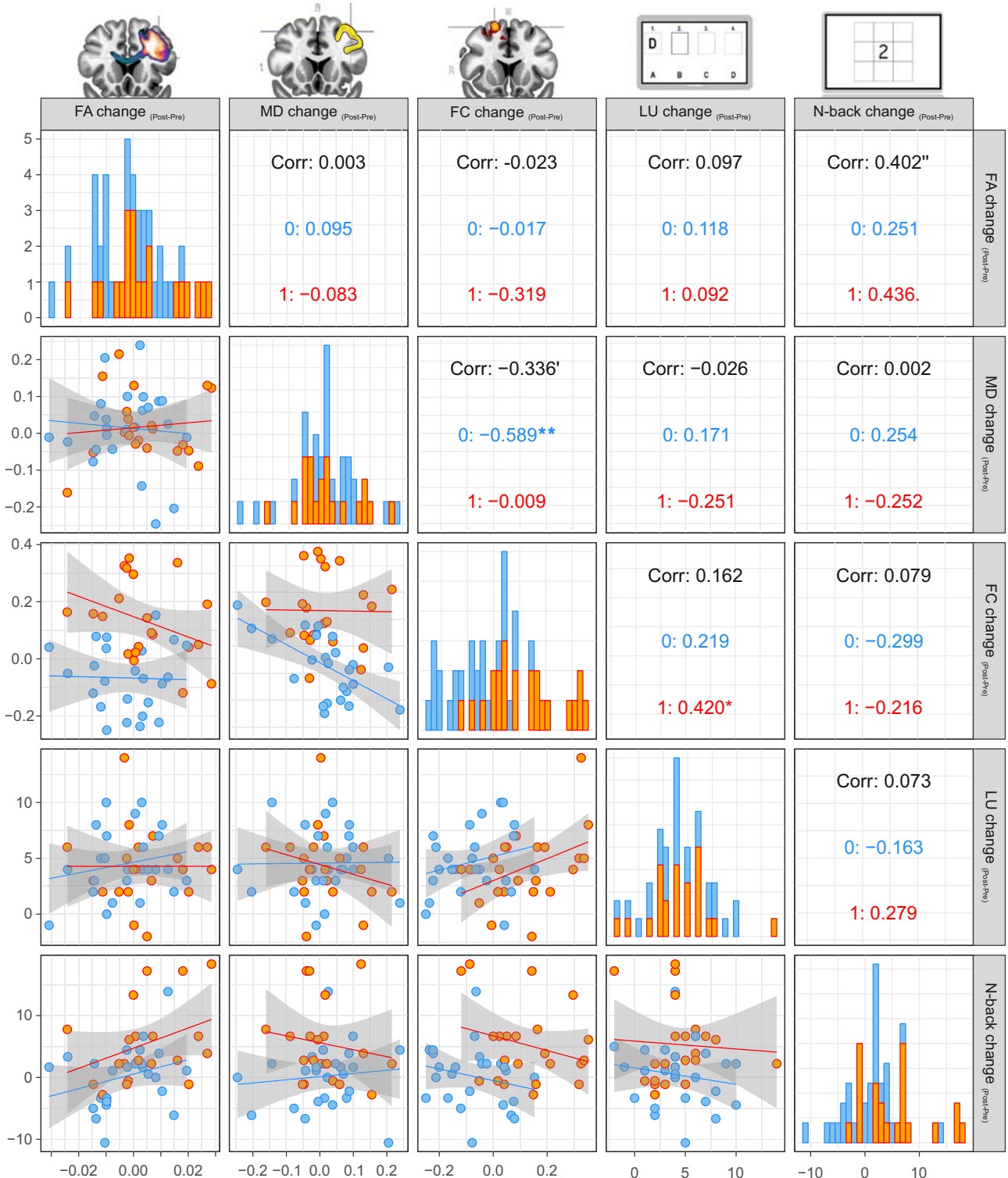

**Fig. 5 | Scatterplots for correlations between Post-Pre differences in FA, MD, and FC with individual performance gain (LU and N-back change).** Brain images were created with MRIcroGL (https://www.nitrc.org/projects/mricrogl). Spearman's rank correlation coefficents are shown. Two-sided statistical testing was performed and no adjustments for multiple comparisons were made. Increased FA change was associated with more pronounced gain in N-back task. Increased FC change was associated with more pronounced gain in LU task. Decreased MD changes were associated with FC increases. FA fractional anisotropy, MD mean diffusivity, FC functional connectivity, LU letter updating. Blue bars/points/0: sham group. Orange bars/points/1: anodal tDCS group.. $p = 0.054$, *$p = 0.046$, '$p = 0.022$, "$p = 0.006$, **$p = 0.002$. Source data are provided as a Source Data file.

values indicate reduced restriction of water molecule diffusion by cellular structures[17]. Decreases in MD were also related to brain-derived neurotrophic factor increase (BDNF) which is a marker of LTP[19]. Next to the expression of BDNF, increases in number of synapses and higher astocyte activation has been observed and thus discussed as potential underlying mechanisms of learning-induced structural remodeling of neurons and/or glia, sensitive to MD modulation[15,18,19].

Our finding of decreased MD in the anodal compared to sham group may indicate increases in tissue density (due to reshaping of neuronal or glial processes) or enhanced tissue organization (due to strengthened dendrites or axons) due to tDCS[19,69]. In the rat brain, tDCS modulated spinogenesis (increasing the number and affecting the shape of spines) in the auditory cortex, not only inducing the formation of new spines, but also stabilizing already existing connections[70]. We observed a slight, though statistically not different, "numerical" increase of MD values from before to after the combined intervention, similar to what has been found after an exercise training in older adults: Here, Callow and colleagues found increases in cortical gray matter (insular) MD after training, that were associated with better cognitive performance[71]. These training-induced MD increases could be interpreted as reduced cellular swelling in the aged brain or an enhanced neural efficiency through synaptic and dendritic pruning (reducing density of synapses and dendrites and thus increasing MD values)[62,72]. Together with these findings, our results corroborate the preservation of dynamic properties of glial-related activity for the refinement of synaptic processes in aged individuals. TDCS, however, may also operate upon dendritic spine sprouting and branching, synaptogenesis, and/or increases of glial cell volume[15,24].

It is important to note that DTI metrics are only indirect measures of microstructure[16]. For MD changes, cumulative evidence suggests that the directionality (i.e., increase vs. decrease) and its interpretation might depend on the targeted brain structure, participant group (i.e., physiological or pathological condition), and the specific interventional approach under study[71,73]. Differences in inflammation and hydration/edema also contribute to MD paramaters[62,74,75]. For instance, MD values were elevated in acute multiple sclerosis lesions[74–76], known to involve inflammatory processes (also reflected in additional MR parameters like gadolinium enhancement)[17,77]. Such inflammatory changes have not been observed after tDCS[78]; thus, MD changes in healthy older adults induced by an atDCS-plus-training intervention are unlikely to be due to inflammatory processes (note, however, that there is some preliminary evidence for modulation of neuroinflammatory response through cathodal tDCS in experimental models of epilepsy[79] and stroke[80]). Future studies that combine several neuroimaging measures (such as perfusion, spectroscopy, etc.) may allow to disentangle the exact mechanisms underlying the observed effects[63,74].

In our data, regional MD modulation was not related to performance gain, suggesting a more complex relationship with potentially other influencing factors, such as general training ability[38] or an impact of baseline integrity[81,82]. The lack of a relationship may also point towards an independency of the effects on different modalities, probablity indicating different time scales for the specific level of changes[20,54]. Our findings do not support the hypothesis that tDCS-induced changes in task performance are dependent on changes in regional microstructural integrity itself. However, MD decreases were related to concomitant functional connectivity modulation through training, a finding that further stresses the impact of structural plasticity on brain network connectivity[21,83]. This also highlights the usefulness of multimodal imaging, including comprehensive examination of both gray and white matter plasticity, to uncover the relationships of different levels of effects[25].

In order to examine potential functional connectivity modulations, we conducted seed-based FC analyses using resting-state fMRI. We found increases in FC in the prefrontal task-independent frontoparietal network in the anodal compared to the sham group. Similar FC modulations have been observed in task fMRI during single tDCS applications and after repeated tDCS sessions combined with working memory training in older adults[37,51]. Nissim and colleagues found state-dependent FC increases due to tDCS-accompanied working memory training, within the targeted frontoparietal network[37]. Enhancement of frontoparietal connectivity has been shown to support working memory processing and capacity[84,85]. We previously examined the neural effects of repeated combined tDCS-plus-training sessions such as visuospatial memory. Memory network connectivity was shown to be increased in the tDCS compared to the sham group, indicating coherent intranetwork activity to underlie memory function[4]. Corroborating and extending these and previous findings, we here showed that the functional coupling between bilateral prefrontal regions—part of the frontoparietal network—was increased through tDCS, suggesting modulation of synchronization in neural networks targeted by tDCS as one of the mechanisms underlying tDCS effects. By enabling more coordinated/synchronized activity between network hubs, tDCS combined with repeated sessions may produce (potentially longer-term) transfer effects[2,25,41,86]. In our data, we did not observe an association between FC changes and behavioral performance gains in the transfer task. A lack of a linear association may point towards complexity of the relationship with other influencing factors (such as the impact of baseline FC on behavioral modulation[81,82]), or may be explained by unspecific tDCS effects on different brain areas not neccesarily involved in the task[87]. Previous evidence demonstrated that tDCS can induce network-level changes beyond the stimulation site, demonstrated both in cross-sectional and longitudinal studies[34,35,37,52,55]. In addition, it has been debated that especially the interaction with a particular ongoing task activity may enhance the specificity of tDCS effects[2]. In fact, we observed an association of FC changes with behavioral performance gains in the trained task itself, which is more directly linked to the actual brain stimulation intervention (i.e., task networks directly targeted by tDCS). This link underscores the particular relevance of tDCS-induced functional network alterations for ongoing task activity[2].

A limitation of our study is the relatively small sample size. In particular, in the context of brain-behavior associations, large sample sizes may be required for the observed relationships to be reliable/reproducible[88]. However, neuroimaging data from interventional studies most likely produce larger effect sizes using carefully designed paradigms and measuring well-characterized cognitive processes[89,90], and importantly, allow to establish causal links between human brain and behavior[88]. Therefore, despite the small sample size, repliability is not necessarily limited[89,91]. Given the exploratory nature of our correlational approach to delineate links between levels of neural modulation and behavioral gains through tDCS-plus-training, our findings—while requiring replication—open the path to developing hypotheses for future tDCS studies interrogating specific brain-behavior relationships.

In sum, the present study advances the understanding of neurobiological after-effects of non-invasive brain stimulation combined with repeated training interventions and shows that tDCS exerts its effects on multiple levels, including microstructural properties of white matter tracts and gray matter regions and coordinated activity between distant brain regions. This rapid remodeling of neuroglial networks and long-range signaling as the result of neuromodulation may underlie the functional effects, as indicated by their (partial) association with the observed performance gains[22,25]. Our findings encourage future studies to assess the dynamic properties of microstructural alterations in the human brain in more detail, administering DTI scans within shorter time frames with regard to tDCS-assisted learning. In addition to time scale of remodeling, regional differences remain to be explored in future studies, determining whether neuromodulation excerts similar modulation when applied to other networks. Moreover, it is unclear if findings from the present cohort extend to other (patient) samples as neuromodulatory plasticity may differ as a function of brain changes in different diseases[24,25]. Insights from future investigations will further increase knowledge at microstructural and brain network levels and determinants of responsiveness to stimulation. In a subsequent step, this knowledge may help to develop longer-lasting effects, and potentially to individualize

stimulation parameters including optimal positioning of electrodes and stimulation intensity, in order to maximize functional benefits in experimental and clinical applications[1].

## Methods

### Participants

We performed a monocenter, single-blind randomized, placebo-controlled trial comparing cognitive training to concurrent anodal tDCS (target intervention) with cognitive training to concurrent sham tDCS (control intervention), registered at ClinicalTrial.gov (Identifier NCT03838211). The primary outcome (performance in trained task) and secondary behavioral outcomes (performance on transfer tasks) were reported elsewhere. Here, underlying mechanisms were addressed by pre-specified analyses of multimodal magnetic resonance imaging before and after a three-week executive function training with prefrontal excitatory tDCS in 48 older adults (see Table 1 for demographic characteristics). All participants were right-handed, native German speakers, had no history of neurological or severe psychiatric diseases, did not take any prescription medications (such as antipsychotics, antidepressants, antiepileptics, sedatives, opioids; over the counter medication such as anti-inflammatory drugs like aspirine were allowed), and performed within age- and education-adjusted normative range in the neuropsychological screening (Consortium to Establish a Registry for Alzheimer's Disease, CERAD-Plus Test Battery, https://www.memoryclinic.ch). Inclusion threshold in the CERAD-Plus Test Battery score was defined as performance of each subtest within −1.5 SD from the normative samples' mean[42]. At the screening visit, a total of 14 participants did not meet inclusion criteria and therefore were not invited to participate in the study (of those, $n = 9$ were excluded because of their performance on the CERAD-Plus). Invited participants completed the TrainStim-Cog clinical study where they received anodal or sham transcranial direct current stimulation over the left prefrontal cortex during three weeks (three times a week, totaling up to nine sessions; only one participant missed one session due to sickness) of a training of two executive functions tasks (a letter updating task and a value-based Markov decision making task[41,42], NCT03838211, https://clinicaltrials.gov/show/NCT03838211).

All behavioral data is reported in ref. 41. Sample power calculations were published in the study protocol[42]. Estimating an effect size of 0.8, to demonstrate an effect in the primary outcome, 46 participants (23 per group) had to be included in the analysis with an independent $t$-test using a two-sided significance level of 0.05 and a power of 80 %. Due to an estimated drop-out rate of about 20%, 28 participants were included in each group. Participation in the MRI assessments was not mandatory for inclusion in the trial[42]. Out of 51 participants, 3 ($n = 2$ in anodal group and $n = 1$ in sham group) did not participate in MRI sessions (due to contraindications such as metal in the body or claustrophobia), which resulted in the reduced dataset of $n = 48$. In the present study, we analyzed the magnetic resonance (MR) imaging data acquired in these 48 participants—including resting-state functional MR imaging, structural T1 imaging, and diffusion tensor imaging (DTI)—before and immediately after the three-week intervention. The study flow chart is shown in Fig. 1. The study was approved by the ethics committee of the University Medicine Greifswald and conducted in accordance with the Helsinki Declaration. All participants gave written informed consent before participation.

### Cognitive training with concurrent tDCS

Cognitive training consisted of a letter updating task[68] and a three-stage Markov decision-making task[92]. The tasks were programmed using Unity, C++, Visual basics.NET 15 (letter updating task) and E-prime 3.0 (Markov decision-making task). In the letter updating task, different lists of letters A to D with varying length were presented in random order. After each list, participants were asked to recall the last four letters presented. For the Markov decision-making 3D characters were presented, prompting participants to choose between two actions, which resulted in an action-related outcome (either in terms of monetary gain or loss). Hence, participants had to learn to choose the optimal sequence of actions to maximize their overall gains and minimize overall losses. A numerical n-back-task comprised of a 1-back and a 2-back condition, was used to assess transfer to an untrained working memory task. Each condition consisted of nine trials and 10 items. The task was applied at the session before training (pre) and after training (post). All details including other cognitive tasks are described in refs. 41,42. Cognitive training was accompanied with either anodal or sham tDCS via a battery-operated stimulator (Neuroelectrics Starstim-Home Research Kit). Two circular saline-soaked sponge electrodes (5-cm diameter; anode: F3, cathode: Fp2) mounted in a neoprene head cap were applied using the 10–20 EEG-system grid. Direct current was delivered with 1 mA intensity for 20 minutes in the anodal tDCS group and for 30 s in the sham group. Simulation analyses of the electric field[93] on a Montreal Neurological Institute (MNI) template brain were conducted in order to illustrate that the current reaching the target is well within the range of field strengths assumed to induce neurophysiological effects[94,95] (Supplementary Fig. 5). The stimulation was started simultaneously with the letter updating task (and finished after approximately the first half of the Markov task). Adverse events were assessed by questionnaire every third training session[42,96].

### MRI data acquisition

MR images were acquired at the Baltic Imaging Center (Center for Diagnostic Radiology and Neuroradiology, University Medicine Greifswald) on a 3-T Siemens verio scanner (SIEMENS MAGNETOM Verio syngo MR B17) using a 32-channel head coil. Resting-state fMRI scans were acquired using an echo-planar-imaging sequence ($3 \times 3 \times 3$ mm³ voxel size, repetition time (TR) = 2000 ms, echo time (TE) = 30 ms, flip angle = 90°, 34 slices, descending acquisition, field of view $192 \times 192$ mm², 176 volumes, TA = 6.00 min). Participants were instructed to keep their eyes closed, to not think of anything in particular, and to try not to fall asleep (whether participants fell asleep or not was assessed per self-report directly after the resting-state scan; no participant reported to have fallen asleep). High-resolution anatomical images were acquired using a three-dimensional T1- weighted magnetization prepared rapid gradient echo imaging (1 mm³ isotropic voxel, TR = 2300 ms, TE = 2.96 ms, inversion time = 900 ms, flip angle = 9°, $256 \times 240 \times 192$ mm³ matrix). Further, a diffusion-weighted spin-echo echo-planar imaging sequence was acquired ($1.8 \times 1.8 \times 2.0$ mm³ voxel size, TR = 11100 ms, TE = 107 ms, 70 slices, 64 directions ($b = 1000$ s/mm²), 1 b0).

### MRI data analyses

**Structural T1-weighted images and DTI analysis.** T1 and DTI data were processed by Freesurfer version 6 (https://surfer.nmr.mgh.harvard.edu)[48] and FSL version 6 (https://fsl.fmrib.ox.ac.uk/fsl/fslwiki)[45]. First, T1 data were processed by the FreeSurfer's cross-sectional pipeline (recon-all) which includes motion correction, skull stripping, normalization, intensity correction, volumentric segmentation, and cortical surface reconstruction[97]. Second, the longitudinal pipeline was applied in order to create a robust, unbiased which-subject template using robust, inverse consistent registration which increases reliability and statistical power, for the detection of brain structural changes that may occur with intervention[48,98]. Quality assessment involved visual inspection of all processing steps and calculation of anatomical signal-to-noise ratios using Freesurfer QAtools (https://surfer.nmr.mgh.harvard.edu/fswiki/QATools). All structural data were deemed appropriate for analysis.

Regional volumes were extracted for the ROI corresponding to the stimulation target (i.e., left middle frontal gyri from the Desikan-

Killiani atlas[99]) and adjusted for total intracranial volume (ICV) using the residual-method[100,101].

DTI data preprocessing included eddy current and head motion correction using an automated affine registration algorithm. A diffusion tensor model was fitted to the motion-corrected DTI data at each voxel to create individual 3-dimensional FA and MD maps. FSL's BEDPOSTX was used to calculate the distribution of fiber orientations at each brain voxel. We used a seed-based probabilistic approach to track prefrontal white matter fibers.

Probabilistic fiber tracking was conducted with PROBTRACKX2 implemented in FSL; this method repeatedly samples the distribution at each voxel to produce 'streamlines' that connect voxels from selected seed regions. The following parameters were applied: 5000 streamline samples, 0.5 mm step length, curvature threshold = 0.2. The left middle frontal gyrus from the Harvard-Oxford atlas also used for resting-state fMRI analyses (see below), transformed into individual DTI space, multiplied with diffusion maps and binarized, was used as seed regions for the tracts[102,103]. Given the large size and extent of prefrontal streamlines, these paths were thresholded by 10% of the individual tract-specific connection probability to reduce the likelihood of including extraneous tracts[104]. A canonical image of the thresholded probabilistic tract is provided in Fig. 2. To generate the canonical image, individual tracts from all participants were normalized, converted to binary images and then summed (color coding reflects the probability of voxel to be present in 33–100% of the participants). All data were visually inspected for major artifacts before being included in analyses. Fractional anisotropy (FA) was used as our measure of tract integrity, given that earlier studies have indicated it to be a reliable assessment of microstructural integrity of white matter fibers[105]. Mean FA for all streamlines was then calculated by masking the tracts with individual diffusion maps, binarizing to define tract masks, and averaging individual voxel values along the tract which was then entered into statistical analyses.

Individual T1-weighted images were coregistered to the b0 images, using rigid-body transformation. These registrations were used to transform masks of the left stimulation target to the MD maps. To extract MD from the gray matter within the stimulation target, the individually segmented left middle frontal gyrus was masked by the ROI used for seed-based tractography and rsFC analyses, in line with previous studies[100,106].

To evaluate the robustness of FA results, we conducted complementary tract-based spatial statistics (TBSS)[46] and automated global tractography analyses with anatomical priors (using tracts constrained by underlying anatomy, TRACULA)[47] (see Supplementary Methods and Results for more detailed information).

**Resting-state FC analysis.** Resting-state fMRI data were analyzed using Matlab (v2019a) and CONN toolbox version 21 (www.nitrc.org/projects/conn)[49]. Data preprocessing consisted of functional realignment, slice-time correction, structural segmentation and normalization to the MNI template, functional segmentation and normalization, and smoothing using a 6-mm Gaussian kernel. Denoising of the blood oxygenation level-dependent (BOLD) signal from physiological and other sources of noise was performed using the CompCor method, implemented in the toolbox[107]. The residual BOLD time series were then high pass filtered at 0.01 Hz. Intermediate motion thresholds (0.9 mm slice-to-slice movement and global mean signal below 5 SD) were chosen. Mean and maximum framewise displacement as motion quantities in the anodal and sham group are displayed in Table 2. Scrubbing was implemented as part of the CONN preprocessing pipeline through the Artifact detection toolbox (ART, http://www.nitrc.org/projects/artifact_detect/) by regressing noise components for outlier scans from the BOLD signal as part of denoising[108]. Data with number of detected outlier scans exceeding 3 SD of the sample mean number of outlier scans (mean 7, SD 22) were

**Table 2 | Motion parameters of DTI, rsfMRI data, and T1**

|  | Total | Anodal | Sham |
|---|---|---|---|
| **DTI** |  |  |  |
| *pre* |  |  |  |
| translation (mm) | 1.2 (0.11) | 1.2 (0.10) | 1.2 (0.11) |
| rotation (degrees) | 0.0 (0.0) | 0.0 (0.0) | 0.0 (0.0) |
| *post* |  |  |  |
| translation (mm) | 1.2 (0.11) | 1.2 (0.10) | 1.2 (0.12) |
| rotation (degrees) | 0.0 (0.0) | 0.0 (0.0) | 0.0 (0.0) |
| **rsfMRI** |  |  |  |
| mean FD (mm) | 1.0 (0.6) | 1.0 (0.5) | 1.0 (0.6) |
| max FD (mm) | 0.2 (0.1) | 0.2 (0.1) | 0.2 (0.1) |
| **T1** |  |  |  |
| *pre*: anatomical SNR | 20.6 (2.9) | 20.7 (2.7) | 20.6 (3.2) |
| *post*: anatomical SNR | 20.4 (3.0) | 20.7 (2.6) | 20.2 (3.3) |

Mean and SD values are provided.
Source data are provided as a Source Data file.
*rsfMRI* resting-state functional magnetic resonance imaging, *FD* framewise displacement, *DTI* diffusion tensor imaging, *SNR* signal-to-noise ratio.

excluded from further resting-state analyses, resulting in exclusion of one participant with 147 detected outlier scans (corresponding to 42% of acquired volumes). All segmentation, normalization, and registration steps were visually inspected and were deemed appropriate for analysis.

After preprocessing and denoising, first-level (within-subjects) connectivity maps for each participant were entered into whole-brain region analyses. Second-level (between-subjects) general linear analyses were modeled with a 2 (groups: anodal, sham) × 2 (time points: pre, post) design. The interaction between group and time point was assessed to examine whether functional connectivity alterations from pre to post differed between anodal and sham groups. Age and sex were included as covariates. Analyses were corrected for multiple comparisons using a false discovery rate (FDR)-corrected $p$-value of 0.05 at cluster-level (height threshold of uncorrected $p < 0.001$).

### Statistical analyses
To assess the statistical significance of differences in microstructural MRI markers between stimulation conditions, R[109] was used including the packages emmeans[110], tidyverse[111], ggplot2 and GGally[112]. Linear models were calculated for each dependent variable (FA/MD after intervention). Models were adjusted for age, sex, and respective baseline value. Model-based post-hoc comparisons of estimated fixed effects were computed. $T$-values, degrees of freedom and $p$ values are reported in the "Results" section. A two-sided significance level of $\alpha = 0.05$ was used. No multiple comparison adjustment for $p$-value was performed.

### Reporting summary
Further information on research design is available in the Nature Portfolio Reporting Summary linked to this article.

## Data availability
The processed data of this study are available upon request from the corresponding author. The raw data are not publicly available due to potential identifying information that could compromise participant privacy. Source data are provided with the paper, where the relevant data from each figure or table is represented by a single sheet within the source data file (https://github.com/annaelisabethfromm/NCOM_Antonenko_2023). Source data are provided with this paper.

## Code availability

All analyses were performed using the available toolboxes: R version 4.1.2 (http://www.rproject.org/), MATLAB R2019a (https://www.mathworks.com), CONN v21 (https://web.conn-toolbox.org), Free-Surfer Version 6.0.0 for Segmentation (Version 7.2.0. for TRACULA; http://freesurfer.net) and FSL 6.0.0 (https://fsl.fmrib.ox.ac.uk/fsl/fslwiki/). Costumized codes are available on Github (https://github.com/annaelisabethfromm/NCOM_Antonenko_2023).

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

## Acknowledgements

Research reported in this publication was supported by the Bundesministerium für Bildung und Forschung, Grant/Award Number: 01GQ1424A (AF); the Deutsche Forschungsgemeinschaft, Grant/Award Numbers: 327654276 - SFB1315 (A.F.) and INST 292/155-1 FUGG. The funders had no role in the design and conduct of the study; collection, management, analyses, and interpretation of the data; preparation, review, or approval of the manuscript; and decision to submit the manuscript for publication. We thank Robert Malinowski for MR imaging support.

## Author contributions

D.A. and A.F. conceived the study and designed the experiments; F.T. performed the experiments and collected the MR data; D.A. supervised data collection; D.A., F.T., and A.E.F. processed the MR data; D.A. and A.E.F. analyzed the data and prepared the figures; D.A. prepared the brain images. D.A. and U.G. performed statistical data analysis; D.A., M.M., and A.F. interpreted the results and wrote the paper; all authors reviewed the paper.

## Funding

## Competing interests

The authors declare no competing interests.
