## [Peer Review File · Nature Communications]

Microstructural and functional plasticity following repeated brain stimulation during cognitive training in older adultsReviewer #1 (Remarks to the Author):

This study examined the combination of a 3 week long executive function training regimen with prefrontal tDCS on changes in MRI derived DTI and functional connectivity measures in 48 healthy older adults. Noteworthy results included changes observed in prefrontal white matter microstructure that correlated with gains in executive function. Evidence for reduction in diffusion in grey matter, and increased prefrontal functional connectivity were also observed. The authors interpret these results as evidence of tDCS-induced changes in white matter organization, glia-related and synaptic processes in grey matter, and increased synchronization of targeted functional networks. These are interesting findings that are likely to be noticed in the field.

The significance of this work is somewhat diminished by the fact that other papers have been published previously from the same dataset. These showed a lack of significant improvement in executive function after the three week combined training with tDCS, compared with training without tDCS, in the subject group as a whole. It appears that the present findings are the result of the authors looking more deeply into their dataset to find what factors predicted which individual subjects would benefit more from tDCS, and looking for evidence of the mechanistic effects of tDCS. While these are useful for understanding the effects of tDCS, this particular application of tDCS does not appear to be very effective overall. This may relate to the relatively low current strength and duration used (1 mA intensity for 20 minutes). Why was such a low intensity and duration used? This is not described or supported in the paper.

In terms of comparison to established literature, similar analyses have been performed in previous tDCS studies, especially white matter effects of tDCS, some of which the authors cite. However, the combination of specific training tasks, tDCS montage and outcome measures described here, and the correlation of these specific anatomic effects of tDCS with behavioral effects appears to be novel.

In general, the data as described do support the authors' conclusions. However, their claims regarding the mechanisms behind the observed changes in imaging measures might be correct, but there are alternate possibilities as well. For instance, while FA in white matter pathways does likely include intracellular directional coherence of white matter fibers, it can also be related to extracellular water motion. FA has been shown to reflect both intracellular and extracellular water diffusion. Therefore, their interpretation does not consider other (extracellular) changes related to tDCS. In addition, the DM differences found in gray matter, may relate to changes in synaptic processes, but could also have many other causes, as described below.

Related to this, the second paragraph of the introduction, starting "As for learning-related brain plasticity, the brain's microstructure can be modified by learning." suggests that changes in packing density and fiber geometry are caused by learning. This can be interpreted as saying that new axons are grown as a part of the learning process. Is this what the authors want to convey, and if so, how has this been documented in humans or related species?

In addition, it may be that changes in these measures were related to something else, aside from a direct effect of tDCS and training. For instance, could the observed DM effects have been related to inflammation, changes in hydration, or some other factor that might affect DM for these data?

The authors suggest that enhanced temporal coherence of BOLD activity is one of the mechanisms underlying tDCS effects. However, The way this is written, this suggests the effects are directly related to changes in blood flow and oxygenation. However, this is unlikely, especially given that they did not observe an association between FC changes and behavioral performance gains. It's more likely that the authors meant that changes in functional coherence underly tDCS effects, that are measured using BOLD, and not that BOLD is directly involved. But again, the lack of relationship with behavioral effects makes this finding irrelevant to the main focus of the paper.

Some study details are unclear. For instance, one exclusion listed was that subjects were not allowed to take any central nervous system active medication. The authors likely meant no prescription medications, as many common over the counter medications have central nervous system activity, such as pain-killers and so on. Were these excluded as well?

The methods text describes training “totaling up to nine sessions”, this suggests that some subjects may have received less than 9. Did all subjects included in the analysis receive 9 sessions, or did some receive less than 9?

How was dementia defined using the CERAD-Plus Test Battery, what was the exclusion threshold, and how many potential subjects were excluded?

During the resting state fMRI acquisitions, subjects were asked to try not to fall asleep. How was this assessed or verified? Using eyes closed, it may have been nearly impossible to be sure. Collecting with eyes open would have allowed this. Were subjects excluded who went to sleep?

There is some ambiguity regarding how the experiment was performed. There is no mention of how skin sensation or other potentially adverse events were assessed. How was this done, and how many adverse events were there? There is also no mention of whether blinding was successful based on subject ratings. Was there a difference between groups in their sensation ratings or subjects' estimation of stimulation condition?

In conclusion, the finding of differences in measures of brain structure and connectivity with tDCS is very interesting, and may be useful for improving outcomes with cognitive training and other treatments using tDCS. While multiple interpretations of data acquired using non-invasive imaging methods are always possible, the presence of some form of change seems well supported.

Reviewer #2 (Remarks to the Author):

In this article, the authors examined the neural effects of tDCS-paired cognitive training in healthy older adults to gain a better understanding of the mechanisms supporting previously reported beneficial effects of tDCS. The authors hypothesized that administration of anodal tDCS over the left prefrontal cortex, repeated over nine sessions of executive function training, would result in improved microstructural properties in the PFC and the frontoparietal network relative to sham stimulation. Diffusion tensor imaging (DTI) and fMRI were acquired before and after the three-week training intervention. DTI was used to quantify changes in microstructural properties of the white matter pathways and grey matter in the targeted cortical area, and fMRI, changes in functional connectivity within the frontoparietal network. During each training session, participants performed a letter-updating working memory task and a three-stage Markov decision-making task, with 20 minutes of atDCS starting simultaneously with the letter-updating task and ending halfway through the Markov task. At the pre-, post-, and follow-up sessions, two near-transfer tasks and two far-transfer tasks were administered in addition to the two trained tasks; however, only the near-transfer task for the letter-updating task, the N-back task, was focused on in the present analyses. Their analysis of the DTI data revealed increased functional anisotropy (FA) in tracts connecting the stimulation target within the frontoparietal network in the group receiving tDCS relative to sham. This increase in microstructural integrity was found to be associated with greater performance gains in the N-back task following the intervention. Additionally, following the intervention mean diffusivity (MD) values in the grey matter underlying the stimulation site were found to be lower in the atDCS group compared to sham; further suggesting atDCS improved microstructural properties. Analysis of the resting-state fMRI data revealed a significant increase in functional connectivity between the left and right PFC following the training-plus-tDCS intervention, suggesting atDCS results in greater synchronization within the frontoparietal executive control network.

These results help to fill the gap in our understanding of the neurobiological after-effects of repeated cognitive training interventions with concurrent tDCS. The evidence presented here suggests that such interventions lead to enhanced microstructural modifications in the targeted white matter tracts and grey matter as well as increased synchronization of the targeted functional network. The authors' efforts in this regard are commendable, as an understanding of the mechanisms mediating tDCS-induced behavioral benefits is crucial for developing and implementing effective therapeutic applications involving tDCS. However, there are numerous

clarifications and concerns that should be addressed.

Major

1. During each training session in the TrainStim-Cog clinical study, the participants performed a letter-updating working memory task and a Markov decision-making task. At the pre- and post-sessions, in addition to the two trained tasks, participants also performed four transfer tasks; one near-transfer task for each of the trained tasks (i.e. the n-back task was near-transfer for the letter-updating task and the Wiener matrices test (WMT) was the near-transfer for the Markov decision-making task) and two far-transfer tasks. Out of these 6 assessments, a significant treatment effect was only found for the n-back task. The fact that no significant treatment effect was found for either the letter-updating task, which relies upon similar functional processes as the n-back task is an important point that should be addressed during the exploration of the behavioral relevance of the observed tDCS-induced neural alterations. If the observed atDCS-induced enhancement of the integrity of frontoparietal white matter tracts is to be interpreted as behaviorally beneficial, an explanation is warranted for why these beneficial effects are only apparent in the n-back task.

2. On lines 224-227: "Importantly, the positive relationship of microstructural alterations with behavioral performance gain (as indicated by the transfer N-back task) points towards a functional significance of preserved (brain stimulation-related and learning-related) neuromodulatory plasticity." Along the same lines as the above comment, although the positive relationship between microstructural alterations with behavioral performance gain on the N-back task points towards a functional significance of preserved neuromodulatory plasticity, doesn't the fact that the same relationship was not found for the trained letter-updating task make it more difficult to draw conclusions about the functional significance?

3. How did the authors identify their seed region as the middle frontal gyrus? Was it confirmed that the target of the tDCS montage is indeed in this region based on any form of simulation of the electric field?

4. The distribution of pre-intervention FA and MD values in Figs. 2 and 3 suggest that these distributions might not be similar between the anodal and sham groups. Although the authors include pre-intervention values as a covariate in their analysis, interpreting the interaction effect with pre-intervention values (as is the case with MD) might not be straightforward if these distributions are not similar. Can the authors confirm that the distributions are not different? Alternatively, would it help to perform these statistical analyses on post- minus pre-intervention difference values?

5. The authors state that "fibers connected the prefrontal stimulation target..with ipsilateral parietal areas.." The following line suggests that the authors observed increased fractional anisotropy between the stimulation target and the ipsilateral parietal regions, but I could not find evidence for this in the main text or in the supplementary. Is that the case?

6. The authors' interpretation on the lack of association between behavioral gain and MD and FC is weak. A deeper dive into underlying reasons is needed.

7. What was the reason to switch to the bivariate association approach here, as opposed to the general linear model analysis in other parts of the paper given the correlation between MD and FC?

8. The wording in the Discussion section regarding MD increase/decrease should be improved for clarity.

Minor

1. In the introduction, the authors start by saying that there is preliminary evidence for a benefit of behavioral training combined with tES in advanced age and then jump to discussing how we need a better understanding of the underlying mechanisms by which tDCS exerts beneficial effects in aging brains. There are a couple of issues here. Firstly, no citations are provided that show repeated tDCS combined with cognitive training does exert beneficial effects. These should be

included here. Secondly, while it is reasonable to start out talking about preliminary evidence for beneficial effects of tES more broadly, these statements should be followed up by a discussion of the potential benefits of tDCS more specifically. As it currently reads, the introduction seems to be treating tES and tDCS as synonymous.

2. It is unclear whether all 48 participants completed all nine training sessions. On lines 66-67 it is stated that "All participated in three weekly training sessions provided over three weeks (nine sessions total)." However, on lines 294-296 it says "All participants completed the TrainStim-Cog clinical study where they received anodal or sham transcranial direct current stimulation over the left prefrontal cortex during three weeks (totaling up to nine sessions) of a training...".

3. In the present study, there are 22 participants in the stimulation condition and 26 in the sham condition. In Antonenko et al., (2022) there are 24 participants in the stimulation condition and 27 in the sham condition. Why is there this difference? What criteria were used to determine which subjects to include/exclude?

4. On lines 81-82 it states: "We also explored linear relationships between the effects on different MRI markers, and between MRI markers and performance gain in working memory (i.e., N-back task)." Because it is not mentioned anywhere that no performance gains were observed in the letter-updating working memory task, this statement, while true, can give the impression that the study intervention was an unqualified success with regard to enhancing working memory performance.

5. The article would benefit from clarification of the potential interpretations of the MD results (lines 232-255). On the one hand, we have the suggestion that observed MD increases from before to after intervention could be due to reduced inflammation or improved neural efficiency through synaptic and dendritic pruning, similar to the results seen after exercise training (Callow et al., 2021); seeming to suggest that a smaller increase in MD would be indicative of a smaller improvement in neural efficiency. On the other, we have the observed smaller increases in MD in the tDCS group relative to sham being interpreted as reflecting improved microstructural properties, potentially indicating relative increases in tissue density or strengthened dendrites or axons, and this being linked to learning-induced structural remodeling. In line with the findings from the cited Callow et al. (2021) study, this seems to be saying that an increase in MD is associated with beneficial microstructural changes, yet, at the same time, saying that a decrease in MD is associated with beneficial microstructural changes.

Reviewer #3 (Remarks to the Author):

Summary: the paper by Antonenko et al describe the use of micro-structural MRI (sensitized by diffusion MRI) and fMRI to explore brain plasticity following learning in various task under tDCS.

Novelty and importance: there are frequent criticism regarding the effect of tDCS application. Some would note that its effect is negligible while other overestimate its potential. I think the importance of this paper is in the description that tDCS has direct effect on brain structural and functional plasticity.

Although I am positive about publication of this paper in high impact journal such as nature communications, I have several concerns that needs to be addressed. Some of them are minor and refer to phrasing and style, and some are more substantial regarding analysis and statistical procedures

1. Abstract: terminology is a bit vague. What is enhanced prefrontal white matter? Enhanced how? what are micro-structural gray matter reductions?
2. Similar to the abstract, at the end of the introduction: "improve white matter micro-structure" - how micro-structure can be improved?
3. Results lines 97-102 - I would suggest to show this effect with additional approaches such as TBSS to evaluate the robustness of the observation.

4. Results, lines 114 - again - increase in FA does not necessarily reflect improved white matter but rather changes that occurred in the white matter.
5. Results. Figure 2. The outcome of the probabilistic tractography doesn't fall to a specific fiber system (maybe the corpus callosum as seen from the figure). If so - maybe to show that selecting the corpus callosum through deterministic fiber tracking also replicates the results - could increase the validity dramatically.
6. Results, lines 135-136 - following previous comments, MD decrease does not indicate improved micro-structural properties but rather changes in tissue micro-structure following treatment.
7. Figure 5 and related text: please indicate multiple comparison correction strategy if any. Also, brain-behavior correlation studies are often criticized as under sampled suggesting that to achieve robust results thousands of subjects are required (Marek et al). While I am not sure we can put a threshold number from which these correlations are meaningful, it could be that ~25 in a group is too small. The authors should relate to this issue somehow.

Reviewer #1 (Remarks to the Author):

This study examined the combination of a 3 week long executive function training regimen with prefrontal tDCS on changes in MRI derived DTI and functional connectivity measures in 48 healthy older adults. Noteworthy results included changes observed in prefrontal white matter microstructure that correlated with gains in executive function. Evidence for reduction in diffusion in grey matter, and increased prefrontal functional connectivity were also observed. The authors interpret these results as evidence of tDCS-induced changes in white matter organization, glia-related and synaptic processes in grey matter, and increased synchronization of targeted functional networks. These are interesting findings that are likely to be noticed in the field.

The significance of this work is somewhat diminished by the fact that other papers have been published previously from the same dataset. These showed a lack of significant improvement in executive function after the three week combined training with tDCS, compared with training without tDCS, in the subject group as a whole. It appears that the present findings are the result of the authors looking more deeply into their dataset to find what factors predicted which individual subjects would benefit more from tDCS, and looking for evidence of the mechanistic effects of tDCS. While these are useful for understanding the effects of tDCS, this particular application of tDCS does not appear to be very effective overall. This may relate to the relatively low current strength and duration used (1 mA intensity for 20 minutes). Why was such a low intensity and duration used? This is not described or supported in the paper.

In terms of comparison to established literature, similar analyses have been performed in previous tDCS studies, especially white matter effects of tDCS, some of which the authors cite. However, the combination of specific training tasks, tDCS montage and outcome measures described here, and the correlation of these specific anatomic effects of tDCS with behavioral effects appears to be novel.

Authors' response: We thank the reviewer for her/his overall appreciation.

With regard to "dose" selection (i.e., intensity + duration): A safe and commonly used range of tDCS dose is 1-2 mA for up to 30 min ¹. Previous evidence from proof-of-concept studies suggests general efficacy of applying anodal tDCS with 1 mA for 20 min in single and repeated stimulation sessions, in young and in older adults ^{2,3,4}. These studies have reported modulation of behavioral performance as well as functional activity and connectivity. Titration studies systematically comparing different stimulation parameters have shown non-linearity of intensity-dependent neuroplastic effects (higher intensities not necessarily producing superior outcomes) ^{5,6}; several studies have even found superiority of 1-mA intensity and 20-min duration for different electrode montages and task paradigms including the ones used in the current study ^{7,8,9}. For instance, Ehrhardt et al. observed a transfer of training gains to an untrained task paradigm for 1mA (but not 0.7- nor 2-mA) group, applying anodal tDCS during multisession cognitive training with a prefrontal electrode montage ⁹. Therefore, the choice of stimulation paradigm and set-up is supported by results from previous studies.

Given the above described findings, we opted for 1 mA/20 min; but it is acknowledged that findings are still inconclusive regarding the "optimal" (i.e., most efficient with regard to the desired outcome) stimulation dose, particularly in older populations, as detailed in the next paragraph.

With regard to behavioral "efficacy" in older adults: In older adults, less current may reach the brain due to age-related atrophy which reduces electric fields ¹⁰; here, higher intensities may be necessary to induce behavioral changes ^{11,12}. This claim is supported by our observation of a link between higher individual electric field and higher N-back performance change magnitudes ¹³.

Notwithstanding, we argue that a lack of an overall group effect does not necessarily mean that the intervention was not effective but may rather reflect high interindividual variability (i.e., some individuals may benefit while others do not). Indeed, variability in stimulation response is a common finding in NIBS studies ^{3,14}, which highlights the necessity to carefully investigate the underlying neural mechanisms and predictors.

In response to this query, we have now included the rationale of using the given stimulation dose, as well as more information on the main behavioral results of the study (see also comment #1 of reviewer #2).

In general, the data as described do support the authors' conclusions. However, their claims regarding the mechanisms behind the observed changes in imaging measures might be correct, but there are alternate possibilities as well. For instance, while FA in white matter pathways does likely include intracellular directional coherence of white matter fibers, it can also be related to extracellular water motion. FA has been shown to reflect both intracellular and extracellular water diffusion. Therefore, their interpretation does not consider other (extracellular) changes related to tDCS. In addition, the MD differences found in gray matter, may relate to changes in synaptic processes, but could also have many other causes, as described below.

Authors' response: We agree that there may be alternative interpretations of the observed FA effects (for MD effects, see our response below). We have now expanded the respective paragraphs in the discussion of the revised manuscript.

In the discussion (p. 14), it now reads: Candidate cellular mechanisms reflected in FA variations include alterations in cell membrane and fiber density, fiber coherence, axon diameter, myelination, collateral sprouting. While intracellular directional coherence contributes to the FA metric, extracellular properties have been shown to affect the diffusion of water molecules as well^{15, 16, 17, 18}. Given previous evidence, one possibility is that tDCS may affect fiber organization and myelin formation through rapid structural remodeling in white matter pathways originating from the stimulation target^{19, 20}. These myelination changes would then affect the speed of information processing between brain regions, underlying improvements of performance^{21, 22}. Other hypotheses have to be considered though, such as a potential effect of tDCS on tortuosity in the extracellular space, inducing differential changes in volume fractions in experimental groups (affecting water molecule motion and, as a consequence, the FA values)^{15, 23}. Future methodological research is needed to disentangle the contribution of these potential mechanisms to the observed tDCS-induced changes^{15, 24}. Importantly, the positive correlation of microstructural alterations with behavioral performance gain (as indicated by the transfer N-back task) may point towards a functional significance of preserved (brain stimulation-related and learning-related) neuromodulatory plasticity^{25, 26}.

Related to this, the second paragraph of the introduction, starting "As for learning-related brain plasticity, the brain's microstructure can be modified by learning." suggests that changes in packing density and fiber geometry are caused by learning. This can be interpreted as saying that new axons are grown as a part of the learning process. Is this what the authors want to convey, and if so, how has this been documented in humans or related species?

Authors' response: We thank the reviewer for allowing us to rephrase the respective paragraph to clarify our claims and to improve its comprehensibility. In the previous version, we only referred to papers using DTI in learning/training studies (which cannot convey information about new axon growth and only indirectly link changes in parameters to packing density or fiber geometry changes). We have now completed the paragraph in the introduction by referencing further relevant papers, including the information on species for which changes have been documented.

In the introduction (p. 2), it now reads: As for learning-related brain plasticity, previous work has shown that the brain's microstructure can be modified by learning. Seminal work in post-mortem monkey brains showed that learning of a new skill indeed induces generation of denser and more extensive white matter projections^{27, 28}. In-vivo visualization of learning-induced structural plasticity in both animals and humans is possible with diffusion tensor imaging (DTI)^{17, 29}. Main parameters from DTI sensitive to microstructural changes are fractional anisotropy (FA) and mean diffusivity (MD) with FA in white matter pathways reflecting directional coherence of fibers and MD in grey matter reflecting magnitude of water molecule diffusion¹⁵. Complementary histological analyses showed that, at the cellular level, changes in neural and non-neural dependent activity (e.g., synaptogenesis and changes in dendritic spine morphology) and changes in white matter (e.g., variation of axon diameter, myelin, packing density, fiber geometry) contribute to the observed alterations in neuroimaging data^{17, 25, 29, 30}. For instance, using DTI, Scholz et al. showed that skill training over several weeks induced changes in white matter properties in humans, potentially reflecting changes of myelin, or altered packing density²¹. Similar microstructural remodeling processes following learning, documented by changes in DTI parameters, in both white and grey matter structures have been demonstrated in rodent and human brains^{25, 26, 29, 31}. In sum, while microstructural changes, assessed by DTI, have been demonstrated in several studies to result from training, their exact

timescale, the contributing cellular processes, and their relationship to individual learning magnitudes are yet not completely understood^{21, 29}.

In addition, it may be that changes in these measures were related to something else, aside from a direct effect of tDCS and training. For instance, could the observed MD effects have been related to inflammation, changes in hydration, or some other factor that might affect MD for these data?

Authors' response: Indeed, specificity of DTI metrics such as MD is limited¹⁷. We have now included alternate mechanisms in the discussion of the revised manuscript (see also above as well as our response to minor comment #5 of reviewer #2).

In the discussion (p. 15), it now reads: ... Our finding of decreased MD in the anodal compared to sham group may indicate increases in tissue density (due to reshaping of neuronal or glial processes) or enhanced tissue organization (due to strengthened dendrites or axons) due to tDCS^{25, 32}. In the rat brain, tDCS modulated spinogenesis (increasing the number and affecting the shape of spines) in the auditory cortex, not only inducing the formation of new spines, but also stabilizing already existing connections³³. We observed a slight, though statistically not different, "numerical" increase of MD values from before to after the combined intervention, similar to what has been found after an exercise training in older adults: Here, Callow and colleagues found increases in cortical grey matter (insular) MD after training, that were associated with better cognitive performance³⁴. These training-induced MD increases could be interpreted as reduced cellular swelling in the aged brain or an enhanced neural efficiency through synaptic and dendritic pruning (reducing density of synapses and dendrites and thus increasing MD values)^{16, 35}. Together with these findings, our results corroborate the preservation of dynamic properties of glial-related activity for the refinement of synaptic processes in aged individuals. TDCS, however, may also operate upon dendritic spine sprouting and branching, synaptogenesis, and/or increases of glial cell volume^{24, 29}.

It is important to note that DTI metrics are only indirect measures of microstructure¹⁷. For MD changes, cumulative evidence suggests that the directionality (i.e., increase vs. decrease) and its interpretation might depend on the targeted brain structure, participant group (i.e., physiological or pathological condition), and the specific interventional approach under study^{34, 36}. Differences in inflammation and hydration/edema also contribute to MD parameters^{16, 37, 38}. For instance, MD values were elevated in acute multiple sclerosis lesions^{37, 38, 39}, known to involve inflammatory processes (also reflected in additional MR parameters like gadolinium enhancement)^{15, 40}. Such inflammatory changes have not been observed after tDCS⁴¹; thus, MD changes in healthy older adults induced by an atDCS-plus-training intervention are unlikely to be due to inflammatory processes (note, however, that there is some preliminary evidence for modulation of neuroinflammatory response through cathodal tDCS in experimental models of epilepsy⁴² and stroke⁴³). Future studies that combine several neuroimaging measures (such as perfusion, spectroscopy, etc.) may allow to disentangle the exact mechanisms underlying the observed effects^{18, 37}. ...

The authors suggest that enhanced temporal coherence of BOLD activity is one of the mechanisms underlying tDCS effects. However, the way this is written, this suggests the effects are directly related to changes in blood flow and oxygenation. However, this is unlikely, especially given that they did not observe an association between FC changes and behavioral performance gains. It's more likely that the authors meant that changes in functional coherence underly tDCS effects, that are measured using BOLD, and not that BOLD is directly involved. But again, the lack of relationship with behavioral effects makes this finding irrelevant to the main focus of the paper.

Authors' response: We thank the reviewer for pointing out this wording issue. Indeed, we meant to discuss potential effects on neural networks targeted by tDCS and not direct effects on blood flow and/or oxygenation. This statement has now been rephrased. Moreover, we now lay out in more detail why we believe exploration of FC changes as a function of group may contribute to the main focus of the paper; and have conducted additional analyses to underscore our point.

With regard to the link of resting-state connectivity with behavioral performance gains, and the reviewers' question about relevance of these associations for the present report: A lack of a linear association may point towards complexity of the relationship with other influencing factors (such as the impact of baseline FC on behavioral modulation^{44, 45}), or may be explained by unspecific tDCS effects on different brain areas not

necessarily involved in the task ⁴⁶. Brain stimulation may even produce independent effects on different modalities, probability indicating different time scales for the specific level of changes ^{21,47}. In addition, it has been proposed that especially the interaction with a particular ongoing task activity may enhance the specificity of tDCS effects ⁴⁸ (see also our response to comment #6 of R#2).

The main focus of the paper was to delineate neurobiological after-effects of non-invasive brain stimulation combined with repeated training interventions on multiple imaging modalities, additionally exploring the link to behavioral effects.

Previous evidence demonstrated that tDCS can induce network-level changes beyond the stimulation site, demonstrated both in cross-sectional and longitudinal studies ^{49,50,51,52,53}. We agree that the heterogeneous evidence with regard to the link of those changes with behavioral scores, as evidenced also in our results, may put into question how relevant these changes are, ⁵⁴. However, we believe that conflicting reports underscore the need for further investigation of tDCS-plus-training together with neuroimaging techniques such as fMRI and DTI to advance understanding of neurobiological mechanisms.

In our study, we did not find a relationship of behavioral gains in the transfer (N-back) task with FC changes through the intervention. However, we have now included exploration of performance gain on the trained letter-updating (LU) task (in response to comment #1 of reviewer #2) which is more directly linked to the actual brain stimulation intervention (i.e., task networks directly targeted by tDCS). In fact, modulation of resting-state functional connectivity between the seed and right-hemispheric prefrontal areas showed a positive relationship with LU performance gain in the anodal tDCS group ($\rho=0.42$, $p=0.046$; see Fig. R1 in response to comment#1 of R#2 below), further underscoring the particular relevance of tDCS-induced functional network alterations for ongoing brain activity during training.

We now included further elaboration on this topic in the discussion of the revised manuscript (see p. 17).

Some study details are unclear. For instance, one exclusion listed was that subjects were not allowed to take any central nervous system active medication. The authors likely meant no prescription medications, as many common over the counter medications have central nervous system activity, such as pain-killers and so on. Were these excluded as well?

Authors' response: We apologize for the misleading information. Indeed, we meant prescription medications such as antipsychotics, antidepressants, antiepileptics, sedatives, opioids. Over the counter medication such as anti-inflammatory drugs (e.g., aspirine) were allowed. This has now been clarified (p. 18).

The methods text describes training “totaling up to nine sessions”, this suggests that some subjects may have received less than 9. Did all subjects included in the analysis receive 9 sessions, or did some receive less than 9?

Authors' response: All participants were scheduled to receive all 9 sessions (only n=1 missed the 6th training session due to sickness). This information has now been included (p. 18).

How was dementia defined using the CERAD-Plus Test Battery, what was the exclusion threshold, and how many potential subjects were excluded?

Authors' response: Inclusion threshold in the CERAD-Plus Test Battery score was defined as performance of each subtest within -1.5 SD from the normative samples' mean ⁵⁵. At the screening visit, a total of 14 participants did not meet inclusion criteria and therefore were not invited to participate in the study (of those, n=9 were excluded because of their performance on the CERAD-Plus). This information has now been included (p. 18).

During the resting state fMRI acquisitions, subjects were asked to try not to fall asleep. How was this assessed or verified? Using eyes closed, it may have been nearly impossible to be sure. Collecting with eyes open would have allowed this. Were subjects excluded who went to sleep?

Authors' response: Whether subjects fell asleep or not was assessed per self-report directly after the resting-state scan. No subject reported to have fallen asleep. This information has now been included in the revised version of the manuscript (p. 19).

There is some ambiguity regarding how the experiment was performed. There is no mention of how skin sensation or other potentially adverse events were assessed. How was this done, and how many adverse events were there? There is also no mention of whether blinding was successful based on subject ratings. Was there a difference between groups in their sensation ratings or subjects' estimation of stimulation condition?

Authors' response: We apologize for this missing information.

Adverse events were assessed by questionnaire every third training session^{55, 56}. In our previous publication of the behavioral study results, we had reported these adverse events, showing no difference between groups¹³. We now include the adverse events for the subgroup of n=48 of the present study in the revised manuscript.

It now reads (p. 4): Incidence of adverse events did not differ between groups (incidence rate ratio, 0.8, 95% CI 0.4 – 1.9, see Supplementary Material and Table R1).

In Supplementary Material, it reads: Safety outcomes are reported separately as incidences (n, incidence rate with 95%-CI, based on poisson regression models) in total and by intervention group.

Ten adverse events were reported by seven participants in the target (active stimulation) group and 14 adverse events were reported by eight participants in the control (sham tDCS) intervention group. No serious adverse events were reported and no participant terminated participation due to occurrence of adverse events.

Table R1. Self-reported incidence of adverse events (at least moderate symptoms) by group during intervention.

	Total N=48	Target (active stimulation) intervention group n = 22	Control (sham) intervention group n = 26	Incidence rate ratio for group differences (95 % CI)
Observation time in days, mean (SD)	9.0 (0.1)	9.0 (0.2)	9.0 (0)	
Total number of adverse events	24/ 5.6 (3.6-8.1)	10/ 5.1 (2.5-8.9)	14/ 6.0 (3.4-9.7)	0.8 (0.4-1.9)
Itching	7/ 1.6 (0.7-3.1)	4/ 2 (0.6-4.7)	3/ 1.3 (0.3-3.3)	1.6 (0.3-8.0)
Pain	3/ 0.7 (0.2-1.8)	0/ 0	3/ 1.3 (0.3-3.3)	-
Burning	7/ 1.6 (0.7-3.1)	4/ 2 (0.6-4.7)	3/ 1.3 (0.3-3.3)	1.6 (0.3-8.0)
Warmth/heat	0 / -	0/ -	0 / -	-
Metallic/iron taste	1/ 0.2 (0-1)	0/ -	1/ 0.4 (0-1.9)	-
Fatigue	2/ 0.5 (0.1-1.4)	1/ 0.5 (0-2.2)	1/ 0.4 (0-1.9)	1.2 (0-30)
Other	4/ 0.9 (0.3-2.2)	1/ 0.5 (0-2.2)	3/ 1.3 (0.3-3.3)	0.4 (0-3)

Note: Reported values are absolute frequency of the respective AEs / incidence rate per 100 patient days (95 % CI).

With regard to subjects' guessing of stimulation condition: At the end of all training sessions, participants were asked to guess to which treatment group they were assigned at randomization, see Table R2 for an overview of the answers. We computed the James Blinding Index (BI) where a value of 0.5 (ranging from 0: lack of blinding with all answers correct, to 1: lack of blinding with all answers incorrect; 0.5 means half of the answers are correct, half incorrect) represents random guessing in a randomized, clinical study^{57, 58}. The estimate was 0.67 (95%-CI: 0.55 to 0.80), indicating blinding success.

Table R2. Number of participants by group assignment and guess.

Assignment	Response			Total
	Target	Control	DK	
Target	14	3	5	22
Control	10	3	13	26
Total	24	8	16	48

Note: DK denotes "Don't know".

This information has now been included (see p. 4, and Supplementary Tables S1 and S2).

In conclusion, the finding of differences in measures of brain structure and connectivity with tDCS is very interesting, and may be useful for improving outcomes with cognitive training and other treatments using tDCS. While multiple interpretations of data acquired using non-invasive imaging methods are always possible, the presence of some form of change seems well supported.

Authors' response: We thank the reviewer for this overall positive evaluation of our findings.

Reviewer #2 (Remarks to the Author):

In this article, the authors examined the neural effects of tDCS-paired cognitive training in healthy older adults to gain a better understanding of the mechanisms supporting previously reported beneficial effects of tDCS. The authors hypothesized that administration of anodal tDCS over the left prefrontal cortex, repeated over nine sessions of executive function training, would result in improved microstructural properties in the PFC and the frontoparietal network relative to sham stimulation. Diffusion tensor imaging (DTI) and fMRI were acquired before and after the three-week training intervention. DTI was used to quantify changes in microstructural properties of the white matter pathways and grey matter in the targeted cortical area, and fMRI, changes in functional connectivity within the frontoparietal network. During each training session, participants performed a letter-updating working memory task and a three-stage Markov decision-making task, with 20 minutes of atDCS starting simultaneously with the letter-updating task and ending halfway through the Markov task. At the pre-, post-, and follow-up sessions, two near-transfer tasks and two far-transfer tasks were administered in addition to the two trained tasks; however, only the near-transfer task for the letter-updating task, the N-back task, was focused on in the present analyses. Their analysis of the DTI data revealed increased functional anisotropy (FA) in tracts connecting the stimulation target within the frontoparietal network in the group receiving tDCS relative to sham. This increase in microstructural integrity was found to be associated with greater performance gains in the N-back task following the intervention. Additionally, following the intervention mean diffusivity (MD) values in the grey matter underlying the stimulation site were found to be lower in the atDCS group compared to sham; further suggesting atDCS improved microstructural properties. Analysis of the resting-state fMRI data revealed a significant increase in functional connectivity between the left and right PFC following the training-plus-tDCS intervention, suggesting atDCS results in greater synchronization within the frontoparietal executive control network.

These results help to fill the gap in our understanding of the neurobiological after-effects of repeated cognitive training interventions with concurrent tDCS. The evidence presented here suggests that such interventions lead to enhanced microstructural modifications in the targeted white matter tracts and grey matter as well as increased synchronization of the targeted functional network. The authors' efforts in this regard are commendable, as an understanding of the mechanisms mediating tDCS-induced behavioral benefits is crucial for developing and implementing effective therapeutic applications involving tDCS. However, there are numerous clarifications and concerns that should be addressed.

Major

1. During each training session in the TrainStim-Cog clinical study, the participants performed a letter-updating working memory task and a Markov decision-making task. At the pre- and post- sessions, in addition to the two trained tasks, participants also performed four transfer tasks; one near-transfer task for each of the trained tasks (i.e. the n-back task was near-transfer for the letter-updating task and the Wiener matrices test (WMT) was the near-transfer for the Markov decision-making task) and two far-

transfer tasks. Out of these 6 assessments, a significant treatment effect was only found for the n-back task. The fact that no significant treatment effect was found for either the letter-updating task, which relies upon similar functional processes as the n-back task is an important point that should be addressed during the exploration of the behavioral relevance of the observed tDCS-induced neural alterations. If the observed atDCS-induced enhancement of the integrity of frontoparietal white matter tracts is to be interpreted as behaviorally beneficial, an explanation is warranted for why these beneficial effects are only apparent in the n-back task.

Authors' response: We thank the reviewer for allowing us to clarify this important issue.

In the previous version, we had decided to focus on the link of neural alterations to the behavioral measure which has shown group-wise tDCS effects (i.e., N-back task performance). We now acknowledge that performance in the training task (i.e., letter updating) did not show a difference between anodal and sham groups¹³, and have now detailed on these results in the new version:

As suggested, we now also include letter updating performance into the exploration of behavioral relevance of tDCS-induced neural alterations (Fig. R1). Change in transfer, but not training task performance change correlated with FA changes.

Below and in the revised version of the manuscript, we further elaborate on our observations with additional results. Moreover, we have added information on possible reasons regarding the absence of tDCS-induced effects on letter updating performance, and lack of correlation of individual changes in letter updating performance with individual changes in structural brain correlates (pp. 14f).

Figure R1. Scatterplots for correlations between Post-Pre differences in FA, MD, and FC with individual performance gain (LU and N-back change). Increased FA change was associated with more pronounced gain in N-back task. Increased FC change was associated with more pronounced gain in LU task. Decreased MD changes were associated with FC increases. FA, fractional anisotropy. MD, mean diffusivity. FC, functional connectivity. LU, letter updating. Blue bars/points/0: sham group. Orange bars/points/1: anodal tDCS group. . $p < 0.10$ * $p < 0.05$ ** $p < 0.01$.

Regarding behavioral (group-level) effect in LU, but not N-back task: On the group-level, we observed beneficial effects of the intervention in the transfer task (N-back), but not the training task (LU).

First, Post-Pre outcomes of the N-back and letter-updating task differ with regard to repeated training sessions “in-between” (i.e., LU is administered on nine training sessions while the N-back task is not repeated). As such, we have a strong learning effect in both interventional groups (target and control) as both “normal” and tDCS-accompanied learning follow the same profile ⁵⁹.

Second, the reviewer is correct that both tasks (LU, N-back) rely on similar processing (i.e., the updating, a specific executive function) and engage partly the same neural networks ⁶⁰. However, the two tasks also differ, for instance, in terms of content (letters, numbers), experimental procedure, and brain activation patterns ⁶¹. Differences in functional activation patterns may reflect higher demands on executive processing (continuous updating in conjunction of memorizing for temporal order of letters, in case of the LU task) and active comparison operations (N-back task) ^{60, 62}.

Third, a lack of a group effect does not necessarily mean that the intervention was not effective but may rather reflect a high interindividual variability of stimulation response (with some individuals benefitting from the brain stimulation intervention while others do not). As emphasized in our response to query #1 of reviewer #1) variability in stimulation response is a common finding in NIBS studies ^{3, 14}, which highlights the necessity to carefully investigate the underlying neural mechanisms and predictors.

Regarding behavioral relevance of the observed tDCS-induced microstructural alterations: We observed a correlation of higher FA change with higher N-back change, but not with LU change.

The described differences in the tasks (both related to the procedures of administration and involved executive processes) may also affect the relationship between behavioral modulation and neural plasticity. Improvement in the transfer task (N-back) requires continuous updating and comparison of numbers, which may particularly depend on changes in prefrontal WM pathways' microstructure. While there are also studies reporting microstructural changes after training, the most relevant factors determining changes in FA across repeated sessions are yet unknown (for instance, absolute training duration may be more relevant than performance on the trained task) ^{22, 29}. Our data suggest that microstructural plasticity may be particularly relevant for transfer task performance. Dynamic changes in functional network connectivity may be important for direct training effects in tDCS-plus-training interventions.

2. On lines 224-227: “Importantly, the positive relationship of microstructural alterations with behavioral performance gain (as indicated by the transfer N-back task) points towards a functional significance of preserved (brain stimulation-related and learning-related) neuromodulatory plasticity.” Along the same lines as the above comment, although the positive relationship between microstructural alterations with behavioral performance gain on the N-back task points towards a functional significance of preserved neuromodulatory plasticity, doesn't the fact that the same relationship was not found for the trained letter-updating task make it more difficult to draw conclusions about the functional significance?

Authors' response: Please see our response above.

3. How did the authors identify their seed region as the middle frontal gyrus? Was it confirmed that the target of the tDCS montage is indeed in this region based on any form of simulation of the electric field?

Authors' response: The seed region was defined to represent the gyrus below the anodal electrode (picked from the Harvard-Oxford atlas), centered over F3 ⁶³, consistent with other tDCS studies using ROI approaches that demonstrated neural effects ^{47, 49, 50, 52, 64, 65, 66, 67}.

With regard to electric field simulations: The reviewer is correct that electric field simulations are increasingly used for identification of stimulation targets, as well as post-hoc estimations of induced current to add mechanistic explanations of observed effects^{68,69}.

Stimulation target identification is particularly suitable for focal tDCS montages that use multiple electrodes to constrain the current to a target location. Peak field in conventional tDCS setups (as used in the present study) always lies between the two electrodes^{48,70,71,72}.

Regarding post-hoc estimations of induced current we had performed individual electric field simulations for the present dataset¹³, exploring a positive link of induced whole-brain field magnitude (Fig. 6 of the publication) with individual behavioral performance gain in the N-back task. In response to the reviewers' request, we have now simulated the electric field on an MNI brain to illustrate that the current reaching the target is well within the range of field strengths assumed to induce neurophysiological effects (Fig. R2)^{54,73}.

Figure R2. Electric field distribution of the applied stimulation protocol on an MNI brain using SimNibs⁷⁴ anode centered over the left dorsolateral prefrontal cortex (F3, 5-cm diameter, 1 mA) and return (cathode) centered over the contralateral supraorbital region (Fp2, 5-cm diameter, 1 mA). Field magnitude below the anodal electrode: ~0.15 V/m. LH, left hemisphere.

We have now added the rationale behind seed region selection (pp. 6, 9) and the illustration of the electric field distribution induced by the applied conventional montage (Supplementary Figure 5) to the revised version of the manuscript.

4. The distribution of pre-intervention FA and MD values in Figs. 2 and 3 suggest that these distributions might not be similar between the anodal and sham groups. Although the authors include pre-intervention values as a covariate in their analysis, interpreting the interaction effect with pre-intervention values (as is the case with MD) might not be straightforward if these distributions are not similar. Can the authors confirm that the distributions are not different? Alternatively, would it help to perform these statistical analyses on post- minus pre-intervention difference values?

Authors' response: In order to respond to the reviewers' request, we explored the difference of pre-intervention FA and MD values (Tab. R3). We indeed observed some baseline differences in FA values (mean group difference for anodal > sham: 0.009, $\eta^2 = 0.12$ for SMD) which may occur by chance in randomized controlled trials⁷⁵. As the analyses were adjusted for baseline values, they did not affect the observed stimulation effect. Please note, using post- minus pre-intervention difference values as dependent variable (while similarly adjusting for baseline values) yields the *exact* same results (i.e., stimulation effects)⁷⁶. We did not observe baseline difference in MD values, so the interpretation of the interaction effect is unaffected.

Because formal statistical tests *should not* be conducted on baseline values between randomized groups⁷⁷, we decided against inclusion of this analysis in the manuscript, but would consider reporting it in the supplementary material if the reviewer considers it necessary.

	Mean (SD)		Mean group difference (95% CI)	SMD
	Target (anodal), n=21	Control (sham), n=25		
FA	0.348 (0.023)	0.340 (0.025)	0.009 (-0.006-0.023)	0.36
MD	1.12x10 ⁻³ (0.07x10 ⁻³)	1.12x10 ⁻³ (0.09x10 ⁻³)	0.008x10 ⁻³ (-0.06x10 ⁻³ -0.04x10 ⁻³)	0.10

Note: SMD, standardized mean difference. Following the formula $\eta^2 = (d^2 n) / (d^2 n + n - 1)$, where d is Cohen's d and N is the sample size (n=46) a Cohen's d of 0.36 corresponds to a partial η^2 of 0.12.

5. The authors state that "fibers connected the prefrontal stimulation target..with ipsilateral parietal areas.." The following line suggests that the authors observed increased fractional anisotropy between the stimulation target and the ipsilateral parietal regions, but I could not find evidence for this in the main text or in the supplementary. Is that the case?

Authors' response: The statement referenced from the previous version of the discussion referred to the visual exploration of the canonical tract across subjects. In fact, FA was extracted along the whole individual tracts of each subject, which may vary in their specific trajectories (see the now updated supplementary material to include the individual tracts inputted to produce the canonical image). We apologize for the misleading information that has now been clarified.

It now reads (p.x): Canonical images across our group of participants suggested that white matter fibers project from the stimulation target towards ipsilateral parietal and contralateral prefrontal areas⁷⁸, showing individual differences in their specific trajectories.

6. The authors' interpretation on the lack of association between behavioral gain and MD and FC is weak. A deeper dive into underlying reasons is needed.

Authors' response: We thank the reviewer for this comment. We now expanded our thoughts on the lack of associations (please also see above our response to reviewer #1s' query with regard to BOLD/FC).

It now reads (pp. 16f): ... In our data, regional MD modulation was not related to performance gain, suggesting a more complex relationship with potentially other influencing factors, such as general training ability⁶⁶ or an impact of baseline integrity^{44, 45}. The lack of a relationship may also point towards an independency of the effects on different modalities, probably indicating different time scales for the specific level of changes^{21, 47}. Our findings do not support the hypothesis that tDCS-induced changes in task performance are dependent on changes in regional microstructural integrity itself. However, MD decreases were related to concomitant functional connectivity modulation through training, a finding that further stresses the impact of structural plasticity on brain network connectivity^{31, 79}. ...

... In our data, we did not observe an association between FC changes and behavioral performance gains in the transfer task. A lack of a linear association may point towards complexity of the relationship with other influencing factors (such as the impact of baseline FC on behavioral modulation^{44, 45}), or may be explained by unspecific tDCS effects on different brain areas not necessarily involved in the task⁴⁶. Previous evidence demonstrated that tDCS can induce network-level changes beyond the stimulation site, demonstrated both in cross-sectional and longitudinal studies^{49, 50, 51, 52, 53}. In addition, it has been debated that especially the interaction with a particular ongoing task activity may enhance the specificity of tDCS effects⁴⁸. In fact, we observed an association of FC changes with behavioral performance gains in the trained task itself, which is more directly linked to the actual brain stimulation intervention (i.e., task networks directly targeted by tDCS). This link underscores the particular relevance of tDCS-induced functional network alterations for ongoing task activity⁴⁸.

7. What was the reason to switch to the bivariate association approach here, as opposed to the general linear model analysis in other parts of the paper given the correlation between MD and FC?

Authors' response: Due to the exploratory nature of our association approach (with the main focus of the paper being the neural effects induced by the tDCS-plus-training intervention), we decided to compute monotonic bivariate correlation analyses (instead of multiple linear models) and provide the scatterplots to visualize all individual data. We were specifically interested in the link between behavioral change(s) and the specific level of neural modulation as well as between the levels of neural modulations per se. These links have not been explored so far; thus, there is little knowledge regarding potential (in)dependency and functional significance of the effects. To which extent behavioral and/or a specific neural marker are altered, and how they are linked, may also depend also on other factors (e.g., baseline structural and functional integrity, baseline behavioral performance, age, etc.), so it remains unclear which and how many variables

would need to be included in multiple linear models. Likewise, different factors may influence tDCS effects per se (and consequently their link to neural markers) ⁸⁰. In sum, while our observed links may advance understanding of relationships between behavioral and neural tDCS effects, they are exploratory but will enable future hypotheses-driven investigations.

In order to respond to the reviewers' comment, we have now also computed possible linear models for the two dependent variables (performance change in N-back and performance change in LU training task). These models included all three levels of neural modulation which were studied (FA change in the tract, MD change in the target, and FC change between the target and the significant right-hemispheric cluster) as independent variables (Tab. R4). For N-back change, FA difference from before to after the intervention still showed a positive relationship, despite inclusion of other the variables ($t_{40}=2.57$, $p=0.009$). For LU change, none of the neural markers showed a relationship (the association with FC change becoming non-significant, $t_{41}=1.38$, $p=0.174$, most probably due to our observation from the scatterplots that it was only present in the anodal group). As the link between the dependent variables is not evident from these models, we computed an additional model including MD change as the independent variable and FA change and FC change as covariates. This model showed a less pronounced relationship of MD and FC change ($t_{42}=-1.78$, $p=0.082$) than the unadjusted bivariate correlation. We now state this in the manuscript text, including these linear models into the supplementary material, but keep the correlational analyses.

Table R4. Linear regression analyses.

	B	SE	t	p
N-back change				
Intercept	2.40	0.94	2.57	0.014
FA change	183.0	67.29	2.72	0.009
MD change	0.12	9.49	0.01	0.990
FC change	2.50	5.83	0.43	0.671
LU change				
Intercept	4.40	0.49	9.10	2.24e-11
FA change	15.60	35.00	0.45	0.658
MD change	-1.74	4.87	-0.36	0.723
FC change	4.20	3.04	1.38	0.174
MD change				
Intercept	0.022	0.01	1.48	0.147
FA change	-0.022	1.11	-0.02	0.984
FC change	-0.17	0.09	-1.79	0.082

We have further underlined the exploratory nature of our association approach.

8. The wording in the Discussion section regarding MD increase/decrease should be improved for clarity.

Authors' response: Wording has been improved for clarity (see also below).

Minor

1. In the introduction, the authors start by saying that there is preliminary evidence for a benefit of behavioral training combined with tES in advanced age and then jump to discussing how we need a better understanding of the underlying mechanisms by which tDCS exerts beneficial effects in aging brains. There are a couple of issues here. Firstly, no citations are provided that show repeated tDCS combined with cognitive training does exert beneficial effects. These should be included here. Secondly, while it is reasonable to start out talking about preliminary evidence for beneficial effects of tES more broadly, these statements should be followed up by a discussion of the potential benefits of tDCS more specifically. As it currently reads, the introduction seems to be treating tES and tDCS as synonymous.

Authors' response: We now provide citations that show repeated tDCS combined with cognitive training does exert beneficial effects, e.g.: ^{2,3}, and follow up statements for beneficial effects of tES by a discussion of more specific potential benefits.

It now reads (p. 2): ... Preliminary evidence suggests that the combination of behavioral training and concurrent transcranial electrical stimulation (tES), one of the most widely used non-invasive brain stimulation (NIBS) techniques, may induce cross-task cognitive benefits, in young adults and advanced age ^{2, 3, 48, 81, 82, 83}. In particular, repeated sessions of one variant of tES, anodal transcranial direct current stimulation (tDCS), with cognitive training can boost training gains, with the potential to induce cognitive enhancement lasting up to one month ^{2, 3}. For instance, anodal tDCS over dorsolateral prefrontal cortex during executive training resulted in enhanced working memory performance in anodal compared to sham groups in trained or near-transfer tasks ^{82, 84, 85, 86}. However, evidence on beneficial effects is still not unequivocal ^{87, 88, 89}, and add-on effects are often small and variable between individuals depending on internal or external factors ⁴⁸. Therefore, a better understanding of the underlying mechanisms by which tDCS exerts its beneficial effects in aging brains is of utmost importance to advance the potential of this technique.

2. It is unclear whether all 48 participants completed all nine training sessions. On lines 66-67 it is stated that “All participated in three weekly training sessions provided over three weeks (nine sessions total).” However, on lines 294-296 it says “All participants completed the TrainStim-Cog clinical study where they received anodal or sham transcranial direct current stimulation over the left prefrontal cortex during three weeks (totaling up to nine sessions) of a training...”.

Authors’ response: Only one participant missed one of the training sessions. This information has now been included (p. 18). See also our response to comment 7 of R#1 above.

3. In the present study, there are 22 participants in the stimulation condition and 26 in the sham condition. In Antonenko et al., (2022) there are 24 participants in the stimulation condition and 27 in the sham condition. Why is there this difference? What criteria were used to determine which subjects to include/exclude?

Authors’ response: In this study, participation in the MRI assessments was not mandatory for inclusion in the trial ⁵⁵. Out of 51 participants, 3 (n=2 in anodal group and n=1 in sham group) did not participate in MRI sessions (due to contraindications such as metal in the body or claustrophobia), which resulted in the reduced dataset of n=48. This has now been clarified (p. 18).

4. On lines 81-82 it states: “We also explored linear relationships between the effects on different MRI markers, and between MRI markers and performance gain in working memory (i.e., N-back task).” Because it is not mentioned anywhere that no performance gains were observed in the letter-updating working memory task, this statement, while true, can give the impression that the study intervention was an unqualified success with regard to enhancing working memory performance.

Authors’ response: Indeed, we did not observe differences between anodal and sham groups in the trained letter-updating performance or any other task except the N-back task ¹³. We have now included this information in the revised manuscript. Please also note that we now included exploration of the link between behavioral gain in the trained (LU), in addition to the corresponding near transfer (N-back) with neural plasticity (see also response to major comment #1 above).

In the introduction (pp. 3f), it now reads: Here, we tested the hypotheses that concurrent anodal prefrontal tDCS administered across repeated cognitive training sessions would improve white matter microstructure in cortical target areas and associated neural networks compared to training with placebo (sham) stimulation. tDCS (1 mA) was administered for 20 min concurrently with two executive function training tasks (letter updating training, decision-making). While there were no between-group differences in the primary outcome (performance on letter-updating), we observed superior near-transfer effects (performance on N-back) in the tDCS group at post-intervention and follow-up, but no other transfer tasks (please see ¹³ for the behavioral results of the study). In the current paper, we used DTI acquired before and after the intervention for individual fiber tractography and quantification of white matter microstructure. Further, DTI allowed us to examine whether microstructural properties in the stimulation target would change due to the intervention as suggested previously ⁶⁶. The investigation of microstructural plasticity markers was complemented by resting-state functional magnetic resonance imaging (rs-fMRI) to analyze

functional synchrony modifications within the targeted (fronto-parietal) network. In order to explore the behavioral relevance of neural alterations, we further performed correlational analyses with LU (training, primary behavioral outcome) and N-back (corresponding near-transfer task with enhanced performance in the target compared to the control intervention)¹³.

The end of the first paragraph of the results (p. 4) now reads: We also explored linear relationships between the effects on different MRI markers, and between MRI markers and performance gain in working memory (i.e., LU and N-back task).

5. The article would benefit from clarification of the potential interpretations of the MD results (lines 232-255). On the one hand, we have the suggestion that observed MD increases from before to after intervention could be due to reduced inflammation or improved neural efficiency through synaptic and dendritic pruning, similar to the results seen after exercise training (Callow et al., 2021); seeming to suggest that a smaller increase in MD would be indicative of a smaller improvement in neural efficiency. On the other, we have the observed smaller increases in MD in the tDCS group relative to sham being interpreted as reflecting improved microstructural properties, potentially indicating relative increases in tissue density or strengthened dendrites or axons, and this being linked to learning-induced structural remodeling. In line with the findings from the cited Callow et al. (2021) study, this seems to be saying that an increase in MD is associated with beneficial microstructural changes, yet, at the same time, saying that a decrease in MD is associated with beneficial microstructural changes.

Authors' response: We thank the reviewer for this comment. In fact, we meant to underscore that the directionality of MD changes, i.e., whether increases (=less barriers) or decreases (=more barriers) are beneficial, is not yet completely understood. In general, in older adults, higher age and mild cognitive impairment (MCI) have been associated with grey matter (GM) MD *increases* (reflective of neuronal shrinking, loss of synapses, and increased glial activity which would be interpreted as *detrimental*)^{90, 91, 92, 93}. However, at the onset of cognitive symptoms (such as in MCI), diffusion could also become restricted (=more barriers) due to cellular swelling and inflammation in response to amyloid deposition (here, MD *decreases* reflective of *detrimental* processes)⁹⁴. With regard to neuromodulatory intervention-induced alterations, Callow et al.³⁴ observed MD *increases* in older adults with and without MCI, associated with (exercise) training-induced cognitive *improvements*. The authors interpreted this increase as *beneficial* (being potentially reflective of synaptic pruning, reduced inflammation or cellular swelling in response to the intervention). Others found MD *decreases* due to training in older adults (being potentially reflective of increased dendritic density/aborization)³⁶.

Thus, from current evidence, it is conceivable that whether to expect increases or decreases in MD (to be either *beneficial* or *detrimental*), may depend on the targeted brain region, participant group, and the specific interventional approach^{34, 36}. The lack of clear directionality also emphasizes the need of additional metrics (either imaging or behavior) to interpret any MD modulation in GM (in fact, our observation of MD *decrease* linked to FC increase suggests *beneficial effects*). Please also note, DTI measures can only indirectly assess microstructure with MD (or FA) changes being rather unspecific to particular tissue compartments, given that several anatomical features contribute to these measures^{17, 95}. We have now elaborated more on this topic and clarified potential interpretations of MD results (please see our responses to the related comments of R#1 above).

In the discussion (pp. 15f), it now reads: ... Our finding of decreased MD in the anodal compared to sham group may indicate increases in tissue density (due to reshaping of neuronal or glial processes) or enhanced tissue organization (due to strengthened dendrites or axons) due to tDCS^{25, 32}. In the rat brain, tDCS modulated spinogenesis (increasing the number and affecting the shape of spines) in the auditory cortex, not only inducing the formation of new spines, but also stabilizing already existing connections³³. We observed a slight, though statistically not significant, "numerical" increase of MD values from before to after the combined intervention, similar to what has been found after an exercise training in older adults: Here, Callow and colleagues found increases in cortical grey matter (insular) MD after training, associated with better cognitive performance³⁴. These training-induced MD increases could be interpreted as reduced cellular swelling in the aged brain or an enhanced neural efficiency through synaptic and dendritic pruning (reducing density of synapses and dendrites and thus increasing MD values)^{16, 35}. Together with these

findings, our results corroborate the preservation of dynamic properties of glial-related activity for the refinement of synaptic processes in aged individuals. tDCS, however, may also operate upon dendritic spine sprouting and branching, synaptogenesis, and/or increases of glial cell volume ^{24,29}.

It is important to note that DTI metrics are only indirect measures of microstructure ¹⁷. For MD changes, cumulative evidence suggests that the directionality (i.e., increase vs. decrease) and its interpretation might depend on the targeted brain structure, participant group (i.e., physiological or pathological condition), and the specific interventional approach under study ^{34,36}. Differences in inflammation and hydration/edema also contribute to MD parameters ^{16,37,38}. For instance, MD values were elevated in acute multiple sclerosis lesions ^{37,38,39}, known to involve inflammatory processes (also reflected in additional MR parameters like gadolinium enhancement) ^{15,40}. Such inflammatory changes have not been observed after tDCS ⁴¹; thus, MD changes in healthy older adults induced by an atDCS-plus-training intervention are unlikely to be due to inflammatory processes (note, however, that there is some preliminary evidence for modulation of neuroinflammatory response through cathodal tDCS in experimental models of epilepsy ⁴² and stroke ⁴³). Future studies that combine several neuroimaging measures (such as perfusion, spectroscopy, etc.) may help to improve specificity ^{18,37}. ...

Reviewer #3 (Remarks to the Author):

Summary: the paper by Antonenko et al describe the use of micro-structural MRI (sensitized by diffusion MRI) and fMRI to explore brain plasticity following learning in various task under tDCS.

Novelty and importance: there are frequent criticism regarding the effect of tDCS application. Some would note that its effect is negligible while other overestimate its potential. I think the importance of this paper is in the description that tDCS has direct effect on brain structural and functional plasticity.

Although I am positive about publication of this paper in high impact journal such as nature communications, I have several concerns that needs to be addressed. Some of them are minor and refer to phrasing and style, and some are more substantial regarding analysis and statistical procedures.

1. Abstract: terminology is a bit vague. What is enhanced prefrontal white matter? Enhanced how? what are micro-structural gray matter reductions?
2. Similar to the abstract, at the end of the introduction: ""improve white matter micro-structure" - how micro-structure can be improved?

Authors' response: Has been rephased / wording has been improved (we now refer to "changes" and state "... these changes can be related to/interpreted as enhanced/improved/beneficial ...").

3. Results lines 97-102 - I would suggest to show this effect with additional approaches such as TBSS to evaluate the robustness of the observation.

Authors' response: As suggested, we have performed additional analyses to evaluate the robustness of FA results.

Methods: Voxel-wise statistical analysis of the FA data was carried out using TBSS (Tract-Based Spatial Statistics) ⁹⁶, part of FSL ⁹⁷. Briefly, FSL's nonlinear image registration algorithm was used to align all subjects' FA images to the FMRIB58_FA template in Montreal Neurological Institute (MNI) standard space. A mean skeleton was created which represents the centres of all tracts common to the group. Afterwards each subjects' aligned FA data was projected onto this skeleton (using a lower threshold of 0.2 to include only white matter and reduce the likelihood of partial voluming) and the resulting data was fed into whole-brain voxel-wise cross-subject statistics. The "randomize" algorithm with 5000 permutations was applied, with a cluster significance level of $p < 0.05$ using threshold-free cluster enhancement (TFCE) to control for multiple comparisons over space⁹⁸ (and a minimum cluster size threshold 'k' of 20 voxels). Paired within-subject Post-Pre differences were computed and subjected to a standard general linear model design (similar to our whole-brain voxel-wise statistical design for the resting-state data in CONN). With this approach, a two-sample t-test was performed to test whether the Post-Pre difference differed between

groups (specifically testing the contrast of a relative FA increase in the anodal compared to the sham group), adjusted for age and sex.

Results: A significant relative FA increase in anodal compared to sham group was found in left and right lateral prefrontal, medial prefrontal and parietal regions (permutation test, $p < 0.05$, TFCE-corrected, Tab. R5 and Fig. R3). Cluster sizes and center of gravity cluster MNI coordinates were extracted and regions were labeled with references to John Hopkin University (JHU) white matter (WM) atlas ⁹⁹. Atlas labels mostly corresponded to fiber systems overlapping with the canonical probabilistic pathway (please also see our response below).

Table R5. Clusters of relative FA increase in the anodal compared to sham group.						
Associated cortical regions	Cluster size (mm ³)	Minimum p (TFCE-corrected)	MNI coordinates			Hemi
			x	y	z	
Cingulum/CC	155	0.002	-7.4	5.9	31.1	LH
	21	0.022	8.8	11.1	31.4	RH
SLF	58	0.028	-30.6	-15.2	50.9	LH
	36	0.018	-6.6	-10.9	56.1	LH
	32	0.012	-37.2	-7.7	46.9	LH
	25	0.032	-36.3	14.6	40.1	LH
	25	0.017	-43.0	-56.3	34.3	LH
	24	0.035	-41.1	-4.3	41.0	LH
	20	0.027	-44.9	3.6	16.6	LH
	38	0.036	27.1	3.1	27.2	RH
	31	0.027	49.8	3.4	29.3	RH
	28	0.023	42.7	-3.8	42.9	RH
	27	0.021	31.3	3.4	35.0	RH
ATR	21	0.029	38.1	1.4	25.1	RH
	24	0.012	8.2	-31.4	-10.7	RH
Cerebellum	28	0.018	-11.4	-64.9	-32.2	LH

Note: MNI coordinates (in mm) are given for the center of gravity of the clusters. Neuroanatomical regions were labeled with reference to the John Hopkins University (JHU) atlas. CC, corpus callosum. SLF, superior longitudinal fasciculus. ATR, anterior thalamic radiation. Hemi, hemisphere. LH, left hemisphere. RH, right hemisphere.

Figure R3. Sagittal (top), coronal (middle) and axial (bottom) view of the relative increase of fractional anisotropy (FA) in the anodal compared to sham group obtained with tract-based spatial statistics (TBSS) analysis. The effects were estimated by a whole-brain comparison of anodal and sham group by means of voxel-wise general linear model (GLM, group x time interaction). Red regions (thickened for better visibility) represent tracts with increased FA (permutation test, $p < 0.05$, TFCE-corrected, $k \geq 20$). Results are overlaid on the mean FA skeleton (green) and FSL_HCP1065-FA template (grey). The canonical pathway derived from probabilistic tractography analysis is superimposed (yellow) to illustrate overlap with significant TBSS clusters. On the right: Raincloud plots show FA values averaged over all significant clusters for each group and timepoint. LH, left hemisphere.

While the TBSS approach provides superior between-subject registration and is less subject to partial volume effects, focusing on the center of white matter tracts may not fully capture inter-subject variability¹⁰⁰. Given our confirmatory TBSS results, we are confident that our probabilistic tractography findings are quite robust. We have now highlighted the correspondence of the results obtained with the two approaches (TBSS and tracts from automated global tractography, see below) and included the additional TBSS analysis in the Supplementary Material of the revised manuscript.

4. Results, lines 114 - again - increase in FA does not necessarily reflects improved white matter but rather changes that occurred in the white matter.

Authors' response: Thank you, the respective sentence has been rephrased as suggested.

5. Results. Figure 2. The outcome of the probabilistic tractography doesn't falls to a specific fiber system (maybe the corpus callosum as seen from the figure). If so - maybe to show that selecting the corpus callosum through deterministic fiber tracking also replicates the results - could increase the validity dramatically.

Authors' response: In response to the reviewers' request, we first explored our tractography outcome (i.e., the canonical pathway) more closely and overlaid it with atlas labels of the John's Hopkins University (JHU) white matter (WM) atlas (Fig. R4). The reconstructed tracts connect bilateral prefrontal areas, thus overlapping with the prefrontal part of the body of the corpus callosum (CC); and left lateral prefrontal with left parietal areas, thus overlapping with the superior frontal fasciculus (SLF). We then reconstructed these two specific fiber systems (CC, SLF; and extracted the average FA values) for all individuals and timepoints, see below for details.

Figure R4. Overlay of the canonical tract derived from the probabilistic tractography (yellow) and the JHU WM atlas labels (multicolored). SLF, superior longitudinal fasciculus. CC, corpus callosum. LH, left hemisphere.

Methods: To reconstruct specific fiber systems, automated global tractography with anatomical priors was carried out on the preprocessed FA images (processed with the longitudinal pipeline, see Methods Section) using the tracts constrained by underlying anatomy (TRACULA) tool included in FreeSurfer version 7¹⁰¹. Pathways were estimated based on the training subjects' atlas data combined with the individual segmentation data. Analyses were focused on fibers overlapping with our probabilistic pathway, i.e. connections of the left prefrontal/middle frontal areas (Fig. R4): Corpus callosum (CC), prefrontal section of the body (defined based on its cortical terminations in the rostral subdivision of the superior frontal label or in the rostral middle frontal label) and left superior longitudinal fasciculus (SLF), 2nd branch as defined by anatomical literature (with inclusion ROI in the caudal part of the middle frontal gyrus and in the inferior parietal lobe, and mid-sagittal and temporal exclusion ROI)¹⁰¹. A-posteriori probability distributions were estimated consisting of a likelihood term (estimated from the ball-and-stick model of diffusion) and a term including the estimated pathway priors. FA values were calculated by averaging the individual voxel values along the CC and the SLF for each participant and timepoint.

Results: FA values for each participant and timepoint are displayed in Figure R5. FA values were entered into linear model analyses with values post intervention as dependent variables and group as between-subjects factor (including pre intervention values, age, and sex as covariates). FA values in the CC were higher in the anodal compared to the sham group (main effects $t_{40} = -1.96$, $p = 0.058$, partial $\eta^2 = 0.09$) and

an interaction of initial FA values by group was found ($t_{40} = 2.01$, $p = 0.051$, partial $\eta^2 = 0.09$). Thus, beneficial stimulation effects were larger for individuals with higher FA at baseline (e.g., for low baseline values at 25th percentile (0.52), anodal: 0.52 [0.51, 0.53], sham: 0.52 [0.51, 0.53], $p = 0.530$; for high baseline values at 75th percentile (0.57), anodal: 0.57 [0.56, 0.58], sham: 0.56 [0.55, 0.57], $p = 0.089$). FA in the SLF did not change through the intervention ($t_{41} = 0.02$, $p = 0.984$, partial $\eta^2 = 9.9e-06$; model-derived estimated means [CI]: 0.42 [0.41, 0.42] for anodal and 0.42 [0.41, 0.42] for sham group). In sum, we found increased FA values in the CC (prefrontal section of the body) in anodal compared to sham for individuals with higher baseline FA while no difference was observed for the left SLF.

Figure R5. White matter pathways' microstructure (fractional anisotropy, FA) in two specific fiber tracts of interest (Corpus callosum, CC, and left superior longitudinal fasciculus, SLF) reconstructed using FreeSurfer's TRACULA (v7). The display shows the respective tract in a sample subject (overlaid on the individual FA image). FA along the CC was increased after the intervention in anodal compared to sham group for those individuals with initially higher FA in the tract. FA along the SLF did not change through the intervention.

The fibers reconstructed seeding from the individual left middle frontal gyri vary between individuals in terms of their exact projection/termination (see also representation of canonical tract in Fig. 2 of the manuscript). This variability may lead to our observation that the effects were not observed in a specific fiber system per se (i.e., no main effect for either CC or SLF FA). However, our findings from this complementary analysis suggest that the combined tDCS-plus-training effect may be rather promoted by transcallosal rather than ipsilateral fronto-posterior pathways (i.e., interaction with pre FA values for the CC), consistent with what we found for resting-state FC (i.e., altered FC between prefrontal cortices through the intervention).

We have now included the complementary TRACULA analysis in the Supplementary Material of the revised manuscript.

6. Results, lines 135-136 - following previous comments, MD decrease does not indicate improved micro-structural properties but rather changes in tissue micro-structure following treatment.

Authors' response: Thank you, the respective sentence has been rephrased.

7. Figure 5 and related text: please indicate multiple comparison correction strategy if any. Also, brain-behavior correlation studies are often criticized as under sampled suggesting that to achieve robust results thousands of subjects are required (Marek et al). While I am not sure we can put a threshold number from which these correlations are meaningful, it could be that ~25 in a group is too small. The authors should relate to this issue somehow.

Authors' response: We thank the reviewer for raising this point. No multiple correction strategy was applied for these explorative correlational analyses. We now discuss these issues in the text in more detail.

With regard to sample size: We are aware of the ongoing debate about sample sizes to detect reliable brain-behavior associations ^{102, 103, 104}. As the reviewer mentions, according to Marek et al. 2022 ¹⁰⁴, studies investigating brain-wide associations should include large sample sizes (N>1,000; obtained from consortia datasets, for example) for the findings to be reliable/reproducible. However, the authors explicitly mention that this applies for brain-wide associations studies (BWAS) “in contrast to non-BWAS approaches with larger effect sizes (for example, lesions, interventions and within-person)” (Abstract of the paper, last sentence). In fact, “some of the most well-replicated findings in human neuroscience come from studies that used carefully designed task paradigms to measure well-characterized cognitive processes in a small number of individuals” ¹⁰⁵. So while large samples may be particularly important for BWAS, small-sample neuroimaging data from ‘focused studies’ (maximizing signal while minimizing noise, thus increasing the signal-to-noise ratio) ¹⁰³ and non-BWAS approaches such as our interventional study can establish fundamental links between human brain and behavior ¹⁰⁴, or tDCS-induced modulatory neural plasticity with behavioral gain, respectively, as in our specific case. As such, Marek et al. specifically acknowledge that “small-sample neuroimaging will always be critical for studying the human brain” (p. 658, last sentence). Such neuroscientific interventional (brain stimulation) studies produce larger effect sizes and have larger statistical power ^{103, 104, 106}. In sum, while our findings are still exploratory, raising further questions and requiring replication in the future, we believe they contribute valuable conclusions to the field by revealing neuromodulatory plasticity through non-invasive brain stimulation.

We have now expanded on these thoughts in the revised version of the manuscript.

It now reads (p. 17): A limitation of our study is the relatively small sample size. In particular, in the context of brain-behavior associations, large sample sizes may be required for the observed relationships to be reliable/reproducible ¹⁰⁴. However, neuroimaging data from interventional studies most likely produce larger effect sizes using carefully designed paradigms and measuring well-characterized cognitive processes ^{103, 105}, and importantly, allow to establish causal links between human brain and behavior ¹⁰⁴. Therefore, despite the small sample size, reliability is not necessarily limited ^{103, 106}. Given the exploratory nature of our correlational approach to delineate links between levels of neural modulation and behavioral gains through tDCS-plus-training, our findings – while requiring replication – open the path to developing hypotheses for future tDCS studies interrogating specific brain-behavior relationships.

References

1. Grossman P, Woods AJ, Knotkova H, Bikson M. Safety of Transcranial Direct Current Stimulation. In: *Practical Guide to Transcranial Direct Current Stimulation: Principles, Procedures and Applications* (eds Knotkova H, Nitsche MA, Bikson M, Woods AJ). Springer International Publishing (2019).
2. Meinzer M, et al. Transcranial direct current stimulation over multiple days improves learning and maintenance of a novel vocabulary. *Cortex* **50**, 137-147 (2014).
3. Perceval G, Martin AK, Copland DA, Laine M, Meinzer M. Multisession transcranial direct current stimulation facilitates verbal learning and memory consolidation in young and older adults. *Brain Lang* **205**, 104788 (2020).
4. Goldthorpe RA, Rapley JM, Violante IR. A Systematic Review of Non-invasive Brain Stimulation Applications to Memory in Healthy Aging. *Front Neurol* **11**, 575075 (2020).
5. Jamil A, et al. Systematic evaluation of the impact of stimulation intensity on neuroplastic after-effects induced by transcranial direct current stimulation. *J Physiol* **595**, 1273-1288 (2017).
6. Batsikadze G, Moliadze V, Paulus W, Kuo MF, Nitsche MA. Partially non-linear stimulation intensity-dependent effects of direct current stimulation on motor cortex excitability in humans. *J Physiol* **591**, 1987-2000 (2013).

7. Papazova I, *et al.* Improving working memory in schizophrenia: Effects of 1 mA and 2 mA transcranial direct current stimulation to the left DLPFC. *Schizophr Res* **202**, 203-209 (2018).
8. Weller S, Nitsche MA, Plewnia C. Enhancing cognitive control training with transcranial direct current stimulation: a systematic parameter study. *Brain Stimul* **13**, 1358-1369 (2020).
9. Ehrhardt SE, Filmer HL, Wards Y, Mattingley JB, Dux PE. The influence of tDCS intensity on decision-making training and transfer outcomes. *J Neurophysiol* **125**, 385-397 (2021).
10. Antonenko D, Grittner U, Saturnino G, Nierhaus T, Thielscher A, Flöel A. Inter-individual and age-dependent variability in simulated electric fields induced by conventional transcranial electrical stimulation. *NeuroImage* **224**, 117413 (2021).
11. Indahlastari A, *et al.* Modeling transcranial electrical stimulation in the aging brain. *Brain Stimul* **13**, 664-674 (2020).
12. Ghasemian-Shirvan E, *et al.* Optimizing the Effect of tDCS on Motor Sequence Learning in the Elderly. *Brain Sci* **13**, (2023).
13. Antonenko D, *et al.* Randomized trial of cognitive training and brain stimulation in non-demented older adults. *Alzheimer's & dementia (New York, N Y)* **8**, e12262 (2022).
14. Wiethoff S, Hamada M, Rothwell JC. Variability in response to transcranial direct current stimulation of the motor cortex. *Brain Stimul* **7**, 468-475 (2014).
15. Le Bihan D, Johansen-Berg H. Diffusion MRI at 25: exploring brain tissue structure and function. *NeuroImage* **61**, 324-341 (2012).
16. Le Bihan D, *et al.* Diffusion tensor imaging: concepts and applications. *J Magn Reson Imaging* **13**, 534-546 (2001).
17. Assaf Y, Johansen-Berg H, Thiebaut de Schotten M. The role of diffusion MRI in neuroscience. *NMR Biomed* **32**, e3762 (2019).
18. Le Bihan D. Looking into the functional architecture of the brain with diffusion MRI. *Nat Rev Neurosci* **4**, 469-480 (2003).
19. Zheng X, Schlaug G. Structural white matter changes in descending motor tracts correlate with improvements in motor impairment after undergoing a treatment course of tDCS and physical therapy. *Frontiers in human neuroscience* **9**, 229 (2015).
20. Wake H, Lee PR, Fields RD. Control of local protein synthesis and initial events in myelination by action potentials. *Science* **333**, 1647-1651 (2011).
21. Scholz J, Klein MC, Behrens TE, Johansen-Berg H. Training induces changes in white-matter architecture. *Nat Neurosci* **12**, 1370-1371 (2009).
22. Takeuchi H, *et al.* Training of working memory impacts structural connectivity. *J Neurosci* **30**, 3297-3303 (2010).
23. Sullivan EV, Pfefferbaum A. Diffusion tensor imaging and aging. *Neurosci Biobehav Rev* **30**, 749-761 (2006).
24. Barbati SA, Podda MV, Grassi C. Tuning brain networks: The emerging role of transcranial direct current stimulation on structural plasticity. *Front Cell Neurosci* **16**, 945777 (2022).
25. Sagi Y, Tavor I, Hofstetter S, Tzur-Moryosef S, Blumenfeld-Katzir T, Assaf Y. Learning in the fast lane: new insights into neuroplasticity. *Neuron* **73**, 1195-1203 (2012).

26. Hofstetter S, Tavor I, Tzur Moryosef S, Assaf Y. Short-term learning induces white matter plasticity in the fornix. *J Neurosci* **33**, 12844-12850 (2013).
27. Hihara S, *et al.* Extension of corticocortical afferents into the anterior bank of the intraparietal sulcus by tool-use training in adult monkeys. *Neuropsychologia* **44**, 2636-2646 (2006).
28. Johansen-Berg H. Structural plasticity: rewiring the brain. *Curr Biol* **17**, R141-144 (2007).
29. Zatorre RJ, Fields RD, Johansen-Berg H. Plasticity in gray and white: neuroimaging changes in brain structure during learning. *Nat Neurosci* **15**, 528-536 (2012).
30. Blumenfeld-Katzir T, Pasternak O, Dagan M, Assaf Y. Diffusion MRI of structural brain plasticity induced by a learning and memory task. *PLoS One* **6**, e20678 (2011).
31. Brodt S, Gais S, Beck J, Erb M, Scheffler K, Schönauer M. Fast track to the neocortex: A memory engram in the posterior parietal cortex. *Science* **362**, 1045-1048 (2018).
32. Assaf Y, Pasternak O. Diffusion tensor imaging (DTI)-based white matter mapping in brain research: a review. *J Mol Neurosci* **34**, 51-61 (2008).
33. Paciello F, *et al.* Anodal transcranial direct current stimulation affects auditory cortex plasticity in normal-hearing and noise-exposed rats. *Brain Stimul* **11**, 1008-1023 (2018).
34. Callow DD, *et al.* Exercise Training-Related Changes in Cortical Gray Matter Diffusivity and Cognitive Function in Mild Cognitive Impairment and Healthy Older Adults. *Frontiers in aging neuroscience* **13**, 645258 (2021).
35. Assaf Y. Can we use diffusion MRI as a bio-marker of neurodegenerative processes? *Bioessays* **30**, 1235-1245 (2008).
36. Kleemeyer MM, *et al.* Changes in fitness are associated with changes in hippocampal microstructure and hippocampal volume among older adults. *NeuroImage* **131**, 155-161 (2016).
37. Alexander AL, Lee JE, Lazar M, Field AS. Diffusion tensor imaging of the brain. *Neurotherapeutics* **4**, 316-329 (2007).
38. Sykova E, Nicholson C. Diffusion in brain extracellular space. *Physiol Rev* **88**, 1277-1340 (2008).
39. Tievsky AL, Ptak T, Farkas J. Investigation of apparent diffusion coefficient and diffusion tensor anisotropy in acute and chronic multiple sclerosis lesions. *AJNR Am J Neuroradiol* **20**, 1491-1499 (1999).
40. Janve VA, *et al.* The radial diffusivity and magnetization transfer pool size ratio are sensitive markers for demyelination in a rat model of type III multiple sclerosis (MS) lesions. *NeuroImage* **74**, 298-305 (2013).
41. Nitsche MA, *et al.* MRI study of human brain exposed to weak direct current stimulation of the frontal cortex. *Clinical Neurophysiology* **115**, 2419-2423 (2004).
42. Regner GG, *et al.* Transcranial direct current stimulation (tDCS) affects neuroinflammation parameters and behavioral seizure activity in pentylentetrazole-induced kindling in rats. *Neurosci Lett* **735**, 135162 (2020).
43. Zhang KY, *et al.* Cathodal tDCS exerts neuroprotective effect in rat brain after acute ischemic stroke. *BMC Neurosci* **21**, 21 (2020).
44. Allman C, *et al.* Ipsilesional anodal tDCS enhances the functional benefits of rehabilitation in patients after stroke. *Sci Transl Med* **8**, 330re331 (2016).

45. Ghobadi-Azbari P, *et al.* fMRI and transcranial electrical stimulation (tES): A systematic review of parameter space and outcomes. *Progress in neuro-psychopharmacology & biological psychiatry* **107**, 110149 (2021).
46. Leaver AM, *et al.* Modulation of brain networks during MR-compatible transcranial direct current stimulation. *NeuroImage* **250**, 118874 (2022).
47. Bachtiar V, Near J, Johansen-Berg H, Stagg CJ. Modulation of GABA and resting state functional connectivity by transcranial direct current stimulation. *Elife* **4**, e08789 (2015).
48. Polanía R, Nitsche MA, Ruff CC. Studying and modifying brain function with non-invasive brain stimulation. *Nat Neurosci* **21**, 174-187 (2018).
49. Meinzer M, *et al.* Electrical brain stimulation improves cognitive performance by modulating functional connectivity and task-specific activation. *J Neurosci* **32**, 1859-1866 (2012).
50. Meinzer M, Lindenberg R, Antonenko D, Flaisch T, Flöel A. Anodal transcranial direct current stimulation temporarily reverses age-associated cognitive decline and functional brain activity changes. *J Neurosci* **33**, 12470-12478 (2013).
51. Keeser D, *et al.* Prefrontal transcranial direct current stimulation changes connectivity of resting-state networks during fMRI. *J Neurosci* **31**, 15284-15293 (2011).
52. Stagg CJ, *et al.* Widespread Modulation of Cerebral Perfusion Induced during and after Transcranial Direct Current Stimulation Applied to the Left Dorsolateral Prefrontal Cortex. *Journal of Neuroscience* **33**, 11425-11431 (2013).
53. Nissim NR, *et al.* Effects of Transcranial Direct Current Stimulation Paired With Cognitive Training on Functional Connectivity of the Working Memory Network in Older Adults. *Frontiers in aging neuroscience* **11**, 340 (2019).
54. Filmer HL, Mattingley JB, Dux PE. Modulating brain activity and behaviour with tDCS: Rumours of its death have been greatly exaggerated. *Cortex* **123**, 141-151 (2020).
55. Antonenko D, *et al.* Effects of a Multi-Session Cognitive Training Combined With Brain Stimulation (TrainStim-Cog) on Age-Associated Cognitive Decline - Study Protocol for a Randomized Controlled Phase IIb (Monocenter) Trial. *Frontiers in aging neuroscience* **11**, 200 (2019).
56. Antal A, *et al.* Low intensity transcranial electric stimulation: Safety, ethical, legal regulatory and application guidelines. *Clin Neurophysiol* **128**, 1774-1809 (2017).
57. Arroyo-Fernandez R, Avendano-Coy J, Velasco-Velasco R, Palomo-Carrion R, Bravo-Esteban E, Ferri-Morales A. A New Approach to Assess Blinding for Transcranial Direct Current Stimulation Treatment in Patients with Fibromyalgia. A Randomized Clinical Trial. *Brain Sci* **11**, (2021).
58. Bang H, Ni L, Davis CE. Assessment of blinding in clinical trials. *Control Clin Trials* **25**, 143-156 (2004).
59. Holland R, Crinion J. Can tDCS enhance treatment of aphasia after stroke? *Aphasiology* **26**, 1169-1191 (2012).
60. Dahlin E, Neely AS, Larsson A, Backman L, Nyberg L. Transfer of learning after updating training mediated by the striatum. *Science* **320**, 1510-1512 (2008).
61. Dahlin E, Nyberg L, Backman L, Neely AS. Plasticity of executive functioning in young and older adults: immediate training gains, transfer, and long-term maintenance. *Psychology and aging* **23**, 720-730 (2008).
62. Wager TD, Smith EE. Neuroimaging studies of working memory: a meta-analysis. *Cogn Affect Behav Neurosci* **3**, 255-274 (2003).

63. Koessler L, *et al.* Automated cortical projection of EEG sensors: anatomical correlation via the international 10-10 system. *NeuroImage* **46**, 64-72 (2009).
64. Indahlastari A, *et al.* Individualized tDCS modeling predicts functional connectivity changes within the working memory network in older adults. *Brain Stimul* **14**, 1205-1215 (2021).
65. Nissim NR, *et al.* Effects of in-Scanner Bilateral Frontal tDCS on Functional Connectivity of the Working Memory Network in Older Adults. *Frontiers in aging neuroscience* **11**, 51 (2019).
66. Thams F, Kulzow N, Floel A, Antonenko D. Modulation of network centrality and gray matter microstructure using multi-session brain stimulation and memory training. *Human brain mapping* **43**, 3416-3426 (2022).
67. Antonenko D, *et al.* Age-dependent effects of brain stimulation on network centrality. *NeuroImage* **176**, 71-82 (2018).
68. Hunold A, Haueisen J, Nees F, Moliadze V. Review of individualized current flow modeling studies for transcranial electrical stimulation. *Journal of neuroscience research*, (2022).
69. Evans C, Bachmann C, Lee JSA, Gregoriou E, Ward N, Bestmann S. Dose-controlled tDCS reduces electric field intensity variability at a cortical target site. *Brain Stimul* **13**, 125-136 (2020).
70. Saturnino GB, Madsen KH, Thielscher A. Optimizing the electric field strength in multiple targets for multichannel transcranial electric stimulation. *J Neural Eng* **18**, (2021).
71. Saturnino GB, Antunes A, Thielscher A. On the importance of electrode parameters for shaping electric field patterns generated by tDCS. *NeuroImage* **120**, 25-35 (2015).
72. Datta A, Bansal V, Diaz J, Patel J, Reato D, Bikson M. Gyri-precise head model of transcranial direct current stimulation: improved spatial focality using a ring electrode versus conventional rectangular pad. *Brain Stimul* **2**, 201-207, 207.e201 (2009).
73. Reato D, Rahman A, Bikson M, Parra LC. Low-intensity electrical stimulation affects network dynamics by modulating population rate and spike timing. *J Neurosci* **30**, 15067-15079 (2010).
74. Saturnino GB, Puonti O, Nielsen JD, Antonenko D, Madsen KH, Thielscher A. SimNIBS 2.1: A Comprehensive Pipeline for Individualized Electric Field Modelling for Transcranial Brain Stimulation. In: *Brain and Human Body Modeling: Computational Human Modeling at EMBC 2018* (eds Makarov S, Horner M, Noetscher G). Springer
Copyright 2019, The Author(s). (2019).
75. Altman DG, Doré CJ. Randomisation and baseline comparisons in clinical trials. *Lancet* **335**, 149-153 (1990).
76. Laird N. Further Comparative Analyses of Pretest-Posttest Research Designs. *The American Statistician* **37**, 329-330 (1983).
77. de Boer MR, Waterlander WE, Kuijper LDJ, Steenhuis IHM, Twisk JWR. Testing for baseline differences in randomized controlled trials: an unhealthy research behavior that is hard to eradicate. *International Journal of Behavioral Nutrition and Physical Activity* **12**, 4 (2015).
78. Park HJ, Friston K. Structural and functional brain networks: from connections to cognition. *Science* **342**, 1238411 (2013).
79. Talsma LJ, Kroese HA, Slagter HA. Boosting Cognition: Effects of Multiple-Session Transcranial Direct Current Stimulation on Working Memory. *J Cogn Neurosci* **29**, 755-768 (2017).

80. Koo GKY, *et al.* Identifying factors influencing cognitive outcomes after anodal transcranial direct current stimulation in older adults with and without cognitive impairment: A systematic review. *Neuroscience & Biobehavioral Reviews* **146**, 105047 (2023).
81. Grover S, Wen W, Viswanathan V, Gill CT, Reinhart RMG. Long-lasting, dissociable improvements in working memory and long-term memory in older adults with repetitive neuromodulation. *Nat Neurosci* **25**, 1237-1246 (2022).
82. Ruf SP, Fallgatter AJ, Plewnia C. Augmentation of working memory training by transcranial direct current stimulation (tDCS). *Scientific Reports* **7**, 876 (2017).
83. Antonenko D, Külzow N, Sousa A, Prehn K, Grittner U, Flöel A. Neuronal and behavioral effects of multi-day brain stimulation and memory training. *Neurobiology of aging* **61**, 245-254 (2018).
84. Park S-H, Seo J-H, Kim Y-H, Ko M-H. Long-term effects of transcranial direct current stimulation combined with computer-assisted cognitive training in healthy older adults. *NeuroReport* **25**, (2014).
85. Stephens JA, Berryhill ME. Older Adults Improve on Everyday Tasks after Working Memory Training and Neurostimulation. *Brain Stimulation* **9**, 553-559 (2016).
86. Jones KT, Stephens JA, Alam M, Bikson M, Berryhill ME. Longitudinal Neurostimulation in Older Adults Improves Working Memory. *PLOS ONE* **10**, e0121904 (2015).
87. Nilsson J, Lebedev AV, Rydström A, Lövdén M. Direct-Current Stimulation Does Little to Improve the Outcome of Working Memory Training in Older Adults. *Psychological Science* **28**, 907-920 (2017).
88. Yu J, Lam CLM, Man ISC, Shao R, Lee TMC. Multi-Session Anodal Prefrontal Transcranial Direct Current Stimulation does not Improve Executive Functions among Older Adults. *Journal of the International Neuropsychological Society* **26**, 372-381 (2020).
89. Horne KS, *et al.* Evidence against benefits from cognitive training and transcranial direct current stimulation in healthy older adults. *Nature Human Behaviour* **5**, 146-158 (2021).
90. Zhao X, Wu Q, Chen Y, Song X, Ni H, Ming D. The Conjoint Analysis of Microstructural and Morphological Changes of Gray Matter During Aging. *Front Neurol* **10**, 184 (2019).
91. Pereira JB, *et al.* Regional vulnerability of hippocampal subfields to aging measured by structural and diffusion MRI. *Hippocampus* **24**, 403-414 (2014).
92. Salminen LE, *et al.* Regional age differences in gray matter diffusivity among healthy older adults. *Brain Imaging Behav* **10**, 203-211 (2016).
93. Weston PS, Simpson IJ, Ryan NS, Ourselin S, Fox NC. Diffusion imaging changes in grey matter in Alzheimer's disease: a potential marker of early neurodegeneration. *Alzheimers Res Ther* **7**, 47 (2015).
94. Ryan NS, *et al.* Magnetic resonance imaging evidence for presymptomatic change in thalamus and caudate in familial Alzheimer's disease. *Brain* **136**, 1399-1414 (2013).
95. Johansen-Berg H, Rushworth MF. Using diffusion imaging to study human connective anatomy. *Annu Rev Neurosci* **32**, 75-94 (2009).
96. Smith SM, *et al.* Tract-based spatial statistics: voxelwise analysis of multi-subject diffusion data. *NeuroImage* **31**, 1487-1505 (2006).
97. Smith SM, *et al.* Advances in functional and structural MR image analysis and implementation as FSL. *NeuroImage* **23 Suppl 1**, S208-219 (2004).

98. Smith SM, Nichols TE. Threshold-free cluster enhancement: addressing problems of smoothing, threshold dependence and localisation in cluster inference. *NeuroImage* **44**, 83-98 (2009).
99. Mori S, *et al.* Stereotaxic white matter atlas based on diffusion tensor imaging in an ICBM template. *NeuroImage* **40**, 570-582 (2008).
100. Bennett IJ, Madden DJ. Disconnected aging: cerebral white matter integrity and age-related differences in cognition. *Neuroscience* **276**, 187-205 (2014).
101. Maffei C, *et al.* Using diffusion MRI data acquired with ultra-high gradient strength to improve tractography in routine-quality data. *NeuroImage* **245**, 118706 (2021).
102. Friston K. Sample size and the fallacies of classical inference. *NeuroImage* **81**, 503-504 (2013).
103. Gratton C, Nelson SM, Gordon EM. Brain-behavior correlations: Two paths toward reliability. *Neuron* **110**, 1446-1449 (2022).
104. Marek S, *et al.* Reproducible brain-wide association studies require thousands of individuals. *Nature* **603**, 654-660 (2022).
105. Rosenberg MD, Finn ES. How to establish robust brain-behavior relationships without thousands of individuals. *Nat Neurosci* **25**, 835-837 (2022).
106. Friston K. Ten ironic rules for non-statistical reviewers. *NeuroImage* **61**, 1300-1310 (2012).

Reviewer #1 (Remarks to the Author):

The authors have responded adequately to my previous concerns. I have no additional comments or concerns.

Reviewer #2 (Remarks to the Author):

The authors have addressed all of my concerns satisfactory.

Reviewer #3 (Remarks to the Author):

None